# Cochlear progenitor number is controlled through mesenchymal FGF receptor signaling

Sung-Ho Huh[1]*, Mark E Warchol[2], David M Ornitz[1]*

[1]Department of Developmental Biology, Washington University School of Medicine, St Louis, United States; [2]Department of Otolaryngology, Washington University School of Medicine, St Louis, United States

**Abstract** The sensory and supporting cells (SCs) of the organ of Corti are derived from a limited number of progenitors. The mechanisms that regulate the number of sensory progenitors are not known. Here, we show that Fibroblast Growth Factors (FGF) 9 and 20, which are expressed in the non-sensory (*Fgf9*) and sensory (*Fgf20*) epithelium during otic development, regulate the number of cochlear progenitors. We further demonstrate that *Fgf receptor* (*Fgfr*) *1* signaling within the developing sensory epithelium is required for the differentiation of outer hair cells and SCs, while mesenchymal FGFRs regulate the size of the sensory progenitor population and the overall cochlear length. In addition, ectopic FGFR activation in mesenchyme was sufficient to increase sensory progenitor proliferation and cochlear length. These data define a feedback mechanism, originating from epithelial FGF ligands and mediated through periotic mesenchyme that controls the number of sensory progenitors and the length of the cochlea.

*For correspondence: shuh22@wustl.edu (SH); dornitz@wustl.edu (DMO)

**Competing interests:** The authors declare that no competing interests exist.

## Introduction

The Organ of Corti contains mechanosensory hair cells (HC) and specialized supporting cells (SC) that are required for the transduction of sound (*Wu and Kelley, 2012*). The frequency spectrum of sound stimuli is tonotopically represented along the length of the mammalian cochlea (*Fay and Popper, 2000*). In mouse, the cochlea begins to grow from the ventral otic vesicle at embryonic day 11.5 (E11.5) and continues to grow and coil, forming approximately one and a half turns by birth. During its development, the length of the cochlea is limited by the number of progenitors that give rise to sensory HCs and SCs, and is further regulated through a process of convergent extension (*Chen and Segil, 1999*; *Montcouquiol et al., 2003*; *Wang et al., 2005*; *Wu and Kelley, 2012*). In mouse, sensory progenitors exit the cell cycle by E14.5 and begin to differentiate into HCs and SCs. Thus, the size of the progenitor population at this stage of development is the ultimate determinant of the size of the adult cochlea. Progenitor number is determined by proliferation, the timing of differentiation, and in some cases by aberrant cell death. Previous studies indicate that sensory progenitor growth requires mesenchymal signals (*Phippard et al., 1999*; *Montcouquiol and Kelley, 2003*; *Braunstein et al., 2008*, *2009*), however, the identity and source of the factors that control this activity are not known.

Fibroblast Growth Factors (FGFs) have several stage-specific functions during inner ear development. FGF3 and FGF10 signal from hindbrain and head mesenchyme, respectively, to the overlying ectoderm to induce formation of the otic placode and vesicle (*Urness et al., 2010*). Later in development, FGF20 regulates differentiation of outer hair cells (OHC) and SCs, termed the lateral compartment of the cochlea (*Huh et al., 2012*). Phenotypic similarities with mice lacking *Fgfr1* in the entire otic epithelium suggest that FGF20 signals directly to FGFR1, serving as a permissive factor for differentiation (*Pirvola et al., 2002*; *Hayashi et al., 2008*; *Huh et al., 2012*). FGF9 signaling regulates

**eLife digest** Mammalian ears contain several structures that are involved in hearing. Within the inner ear is a spiral-shaped structure called the cochlea. This contains an array of cells called sensory hair cells that convert sound vibrations into electrical signals, which are then conveyed to the brain. Sounds of differing pitch are detected at different points along the cochlea, so its overall length helps to determine the range of sounds that an individual can hear.

In the embryo, sensory hair cells and their associated supporting cells develop from 'cochlear progenitor' cells. The final length of the cochlea is determined by the numbers of progenitor cells that commit to becoming either sensory hair cells or supporting cells. Two proteins called FGF9 and FGF20 are involved in the formation of the cochlea. FGF20 promotes the formation of the hair cells and supporting cells, but the precise roles of both proteins are not clear.

Here, Huh et al. studied FGF9 and FGF20 in the inner ear of mice at an early stage of development. The experiments show that these proteins work together to control the number of progenitor cells and the length of the cochlea. FGF20 is produced by the same tissue structure (called an 'epithelium') that gives rise to the hair cells and supporting cells. In contrast, FGF9 is produced in another epithelium tissue that produces the cells that line the fluid-filled tubes of the inner ear.

The experiments also show that both FGF9 and FGF20 act as signals to cells in an adjacent tissue called the mesenchyme, where they activate other proteins known as FGF receptors. These receptors, in turn, regulate an unknown molecule in the mesenchyme that influences the growth of progenitor cells and the length of the cochlea.

Sensory hair cells can be injured or lost by excessive sound exposure, some medications and as part of normal aging. These cells are not replaced, and so their loss is a major cause of permanent hearing loss. Understanding the signals that produce the progenitor cells will take us one step closer to being able to grow these cells in the laboratory for use in therapies to replace or repair damaged sensory hair cells.

structural components of the vestibular system, but alone has no effect on cochlear development (*Pirvola et al., 2004*). During postmitotic stages, FGF8 signaling from the inner hair cell (IHC) to FGFR3 in SCs regulates pillar cell differentiation (*Colvin et al., 1996*; *Mueller et al., 2002*; *Jacques et al., 2007*).

Here, we identify another critical stage in inner ear development that requires FGF signaling. We show that *Fgf9*, expressed in the non-sensory epithelium, and *Fgf20*, expressed in the sensory epithelium, regulate the number of cochlear progenitors and the ultimate length of the cochlea through signaling to mesenchymal FGFRs. We find that in vivo FGF9/20 signaling to mesenchymal FGFR1 and FGFR2 is required for sensory progenitor proliferation and that mesenchymal FGFR signaling is sufficient to promote sensory progenitor proliferation and extend the length of the cochlear duct. In addition, we show that prosensory epithelial FGFR1 and FGF20 independently is required for differentiation of outer HCs and SCs.

## Results

### *Fgf9* and *Fgf20* are expressed in the developing cochlea

In a prior study, we showed that *Fgf20* is required between E13.5–14.5 for differentiation of cochlear OHCs and SCs in the organ of Corti (*Huh et al., 2012*). However, *Fgf20* is expressed in a portion of the otic vesicle sensory epithelium much earlier in development, beginning at E10.5 (*Huh et al., 2012*), but analysis of mice lacking *Fgf20* did not reveal any function for *Fgf20* at this stage of development. Since there are many examples of FGFs functioning redundantly during development, we hypothesized that redundancy could account for the lack of a phenotype in *Fgf20* null inner ears between E10.5 and 12.5. *Fgf9* is closely related to *Fgf20* (*Zhang et al., 2006*; *Itoh and Ornitz, 2008*), and is also expressed in the otic epithelium at E10.5–12.5 (*Pirvola et al., 2004*); however, *Fgf9*$^{-/-}$ mice have normal cochlear development and normal patterning of the organ of Corti (*Pirvola et al., 2004*).

We first examined the expression domain of *Fgf9* relative to Sox2-expressing sensory progenitors (and *Fgf20*) using a new *Fgf9*-βGal reporter allele (*Fgf9^lacZ*) in which a splice acceptor-lacZ gene was inserted into the first intron of *Fgf9* (*Skarnes et al., 2011*). At E10.5, βGal activity was detected in the otic vesicle epithelium (*Figure 1A*). Co-staining of βGal and Sox2 at E11.5 showed no overlap, indicating that *Fgf9* is expressed in the non-sensory epithelium of the otic vesicle (*Figure 1B*). Taken together with previous *Fgf20* expression analysis at this stage (*Huh et al., 2012*, *Figure 1C*), *Fgf9* and *Fgf20* are both expressed in the otic vesicle, but in non-overlapping domains in the otic epithelium (*Figure 1D*).

## *Fgf9* and *Fgf20* regulate cochlear length and *Fgf20*, not *Fgf9*, is required for lateral compartment differentiation and patterning

To determine whether *Fgf9* and *Fgf20* could have a redundant role in cochlear development, *Fgf9; Fgf20* double knockout cochleae were analyzed at E18.5 by staining with phalloidin and with an antibody to p75 to identify sensory HCs and pillar cells, respectively (*Figure 2A–C*). Control embryos (*Fgf9^−/+;Fgf20^lacZ/+*) showed a normal pattern of three rows of OHCs and one row of IHCs throughout the cochlear duct (*Figure 2A–C*). *Fgf9^−/−* and *Fgf9^−/−;Fgf20^lacZ/+* cochleae showed the same wild type HC pattern as the double heterozygous controls (*Figure 2A–C*). *Fgf20^lacZ/lacZ* and *Fgf9^−/+;Fgf20^lacZ/lacZ* cochleae showed patches of sensory HCs and gaps (*Figure 2A–C*) (*Huh et al., 2012*). *Fgf9^−/−;Fgf20^lacZ/lacZ* cochleae also showed a similar patterning phenotype to *Fgf20^lacZ/lacZ* mice (*Figure 2A–D*). The density (number of cells per 100 µm) of OHCs in *Fgf9^−/−* and *Fgf9^−/−;Fgf20^lacZ/+* cochleae was similar to double heterozygous controls (*Figure 2E*). However, the densities of OHCs in *Fgf20^lacZ/lacZ*, *Fgf9^−/+;Fgf20^lacZ/lacZ* and *Fgf9^−/−;Fgf20^lacZ/lacZ* cochleae were similar to each other (ANOVA, p > 0.1), and significantly (ANOVA, p < 0.0001) decreased compared to double heterozygous controls (*Figure 2E*). Densities of IHCs were comparable in all genotypes (*Figure 2E*). To analyze SCs, cochleae were immunostained for Prox1 and Sox2 (*Figure 2D*). In double heterozygous control cochleae, 5 rows of Prox1+ SCs overlapped with Sox2 staining (*Figure 2D*). *Fgf9^−/−* and *Fgf9^−/−;Fgf20^lacZ/+* cochleae showed a similar pattern (*Figure 2D*). In contrast, *Fgf20^lacZ/lacZ*, *Fgf9^−/+;Fgf20^lacZ/lacZ* and *Fgf9^−/−;Fgf20^lacZ/lacZ* cochleae showed patches of SCs separated by gaps of Sox2+, Prox1− cells (*Figure 2D*). The density of SCs in *Fgf9^−/−* and *Fgf9^−/−;Fgf20^lacZ/+* was comparable (ANOVA, p > 0.5) to double heterozygous control (*Figure 2F*). The density of SCs in *Fgf20^lacZ/lacZ*, *Fgf9^−/+; Fgf20^lacZ/lacZ* and *Fgf9^−/−;Fgf20^lacZ/lacZ* were similar to each other (ANOVA, p > 0.3) and significantly (ANOVA, p < 0.0001) decreased compared to double heterozygous controls (*Figure 2F*).

One of the striking differences among *Fgf9; Fgf20* compound mutants was cochlear length. The length of *Fgf9^−/−* and *Fgf9^−/−;Fgf20^lacZ/+* cochleae was comparable (ANOVA, p > 0.1) to that of double heterozygous controls (*Figure 2G*), whereas the length of *Fgf20^lacZ/lacZ* and *Fgf9^−/+; Fgf20^lacZ/lacZ* cochleae was 16% and 18% shorter than double heterozygous controls (p < 0.05 and p < 0.001), respectively, and the length of *Fgf9^−/−; Fgf20^lacZ/lacZ* cochleae was reduced by 58% compared to controls (p < 0.001) (*Figure 2G*). In addition, *Fgf9^−/−;Fgf20^lacZ/lacZ* double knockout cochleae were 49% and 51% of the length of *Fgf20^lacZ/lacZ* and *Fgf9^−/+;Fgf20^lacZ/lacZ* cochleae

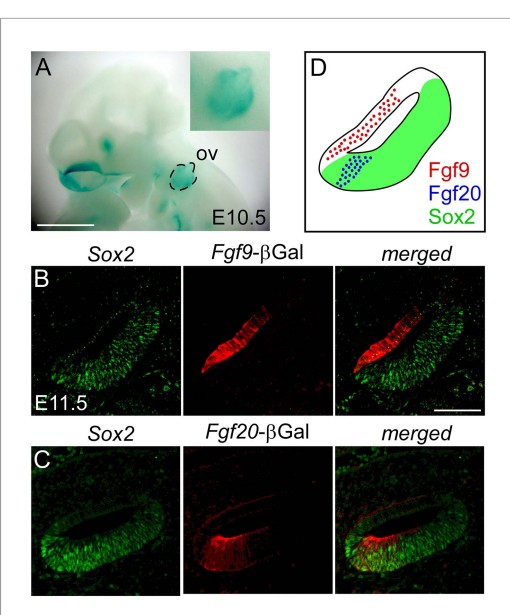

**Figure 1**. *Fgf9* and *Fgf20* are expressed in distinct regions of the otic vesicle. (**A**) βGal activity in an *Fgf9^lacZ/+* embryo at E10.5 visualized with xGal staining. (**B**, **C**) βGal (red) and Sox2 (green) co-immunostaining showing that *Fgf9* (**B**) is expressed in Sox2⁻ non-sensory epithelium and *Fgf20* (**C**) is expressed in Sox2⁺ sensory epithelium at E11.5. (**D**) Schematic diagram of FGF9, FGF20, and Sox2 immunostaining showing that FGF9 and FGF20 are expressed in distinct domains in the otic vesicle. ov, otic vesicle, scale bars, 100 µm.

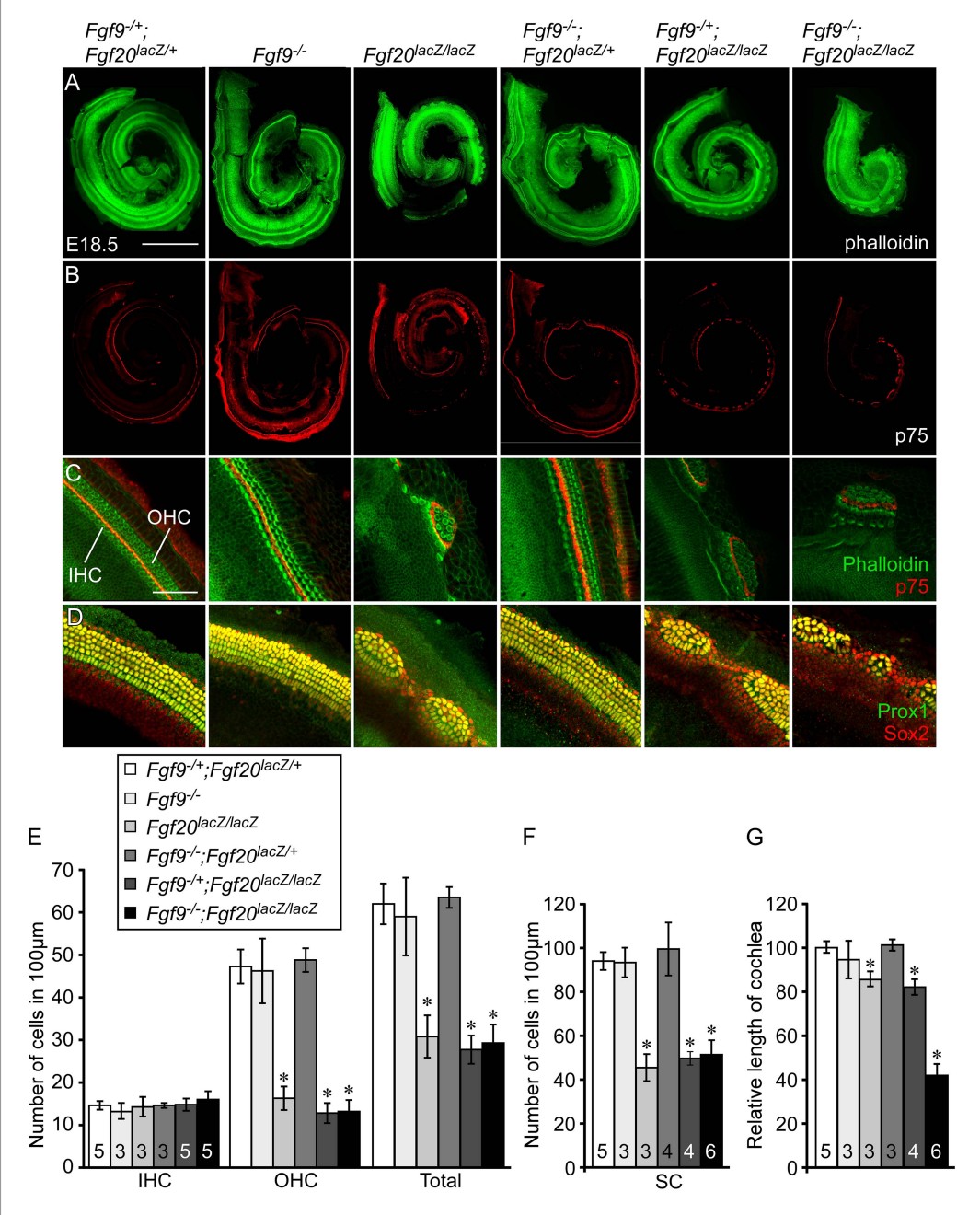

**Figure 2**. *Fgf9* and *Fgf20* regulate cochlear length. (**A**, **B**) Phalloidin (**A**) and p75 immunostaining (**B**) of E18.5 whole cochlea showing hair cells (HCs) (phalloidin) and pillar cells (p75) in the cochlear duct of *Fgf9⁻/⁺;Fgf20^{lacZ/+}*, *Fgf9⁻/⁻*, *Fgf20^{lacZ/lacZ}*, *Fgf9⁻/⁻;Fgf20^{lacZ/+}*, *Fgf9⁻/⁺;Fgf20^{lacZ/lacZ}* and, *Fgf9⁻/⁻;Fgf20^{lacZ/lacZ}* embryos. (**C**) Phalloidin (green) and p75 immunostaining (red) showing the orientation of HCs, pillar cells, and gaps in the sensory epithelium. (**D**) Prox1 (green) and Sox2 (red) co-immunostaining showing supporting cells (SCs) (yellow, Prox1 and Sox2) and undifferentiated sensory progenitors (red, Sox2). (**E–G**) Measurement of HC number (**E**), SC number (**F**), and length of cochleae (**G**) of E18.5 embryos. Scale bars, **A**, 500 μm; **C**, 100 μm. For statistical analysis, all samples were compared with *Fgf9⁻/⁺; Fgf20^{lacZ/+}* double heterozygous controls. *p < 0.001. Sample numbers (n) are indicated in data bars.

(p < 0.001), respectively (*Figure 2G*). These data identify a redundant role for *Fgf9* and *Fgf20* to attain the proper cochlear length, while *Fgf20*, alone, primarily regulates cochlear patterning and differentiation.

## *Fgf9* and *Fgf20* regulate sensory progenitor cells proliferation

We hypothesized that the overall length of the cochlear duct would correlate with the size of the postmitotic prosensory domain. In mouse, cochlear sensory progenitors exit the cell cycle beginning at E12.5 in the apex and progressing towards the base by E14.5 (*Lee et al., 2006*). E14.5 cochleae were dissected and immunostained for Sox2 (*Figure 3A*), which marks the lineage of cells that will become HCs and SCs (*Kiernan et al., 2005*). The Sox2$^+$ prosensory domain of double heterozygous control and *Fgf9*$^{-/+}$;*Fgf20*$^{lacZ/lacZ}$ inner ears were similar (*Figure 3A*). However, the Sox2$^+$ prosensory domain of *Fgf9*$^{-/-}$;*Fgf20*$^{lacZ/lacZ}$ inner ears were clearly smaller than that of double heterozygous control or *Fgf9*$^{-/+}$;*Fgf20*$^{lacZ/lacZ}$ cochleae (*Figure 3A*). Immunostaining of histological sections of E14.5 inner ears showed that the Sox2$^+$ prosensory domain was less compact in *Fgf9*$^{-/-}$;*Fgf20*$^{lacZ/lacZ}$ inner ears compared to double heterozygous control and inner ears with one wild type allele of *Fgf9* (*Figure 3B*). Immunostaining for p27$^{kip1}$ (Cdkn1b) showed a very similar pattern to that of Sox2, with more diffuse cells in the prosensory domain of E14.5 inner ears of *Fgf9*$^{-/-}$;*Fgf20*$^{lacZ/lacZ}$ mice (*Figure 3C*). Jag1, which marks the medial prosensory cells that will give rise to IHCs, inner SCs, and Kölliker's organ (*Ohyama et al., 2010*; *Basch et al., 2011*), showed a similar expression pattern across all three genotypes, indicating that the medial compartment of the cochlea was correctly specified (*Figure 3D*).

To determine whether the decreased size of the *Fgf9*$^{-/-}$;*Fgf20*$^{lacZ/lacZ}$ prosensory domain resulted from changes in cell proliferation and/or cell death, histological sections of E11.5 and E12.5 otic vesicles were immunostained for Sox2 and phospho-Histone H3 (pHH3) (*Figure 3E,F*), or activated Caspase-3 (aCasp3) (data not shown). Quantification of the number of pHH3$^+$, Sox2$^+$ sensory progenitors showed similar numbers ($p > 0.09$ at E11.5 and $p > 0.2$ at E12.5) in double heterozygous control and *Fgf9*$^{-/+}$;*Fgf20*$^{lacZ/lacZ}$ cochleae (*Figure 3G,H*). However, proliferation of *Fgf9*$^{-/-}$;*Fgf20*$^{lacZ/lacZ}$ cochlear epithelial cells was significantly decreased at E11.5 ($p < 0.001$) and E12.5 ($p < 0.01$) compared to double heterozygous controls (*Figure 3G,H*). In addition quantitation of cell proliferation using EdU labeling of E11.5 embryos showed similar results (*Figure 3—figure supplement 1*). No cell death (aCasp3$^+$) was detected in any of the genotypes at E11.5 and E12.5 (data not shown).

Decreased sensory progenitor number could also result from premature cell cycle exit. p27$^{kip1}$ is one of the cell cycle inhibitors that is expressed in sensory progenitors as they become postmitotic. Expression of p27$^{kip1}$ begins at E12.5 in the apex of the cochlea and progresses towards the base (*Lee et al., 2006*). By E14.5, the entire cochlear progenitor domain becomes p27$^{kip1}$ positive. Expression of p27$^{kip1}$ at E12.5 in the proximal cochlear duct was not detected in either control or *Fgf9*$^{-/-}$;*Fgf20*$^{lacZ/lacZ}$ embryos suggesting that there is no premature cell cycle exit in mice lacking *Fgf9* and *Fgf20* (*Figure 3—figure supplement 2*).

## Epithelial *Fgfr1* but not *Fgfr2* is required for lateral compartment differentiation

Next, we questioned which cell types are required for sensory progenitor proliferation and/or lateral compartment differentiation. Expression of both *Fgfr1* and *Fgfr2* have been reported in the otic epithelium and periotic mesenchyme between E10.5 and E12.5 (*Pirvola et al., 2000*, *2002*, *2004*; *Ono et al., 2014*). Epithelial *Fgfr1* has been conditionally inactivated in otic epithelium using *Foxg1*$^{Cre}$, *Six1enh21*$^{Cre}$, and *Emx2*$^{Cre}$ (*Pirvola et al., 2002*; *Ono et al., 2014*). This results in a cochlear epithelium with reduced numbers of HCs, with OHC numbers being more severely affected than IHC numbers. In addition to the loss of differentiated HCs, a 40–50% decrease in cochlear length was reported when *Fgfr1* was inactivated with *Six1enh21*$^{Cre}$, or *Emx2*$^{Cre}$ (*Ono et al., 2014*).

To directly compare cochlear phenotypes resulting from inactivation of *Fgfr1* in otic epithelium with embryos lacking *Fgf9* and *Fgf20*, we re-created and re-evaluated *Fgfr1*$^{-/f}$::*Foxg1*$^{Cre/+}$ mutant mice maintained on a 129X1/SvJ;C57BL/6J mixed genetic background. Quantification of the density of OHCs in *Fgfr1*$^{-/f}$::*Foxg1*$^{Cre/+}$ embryos demonstrated a significant ($p < 0.0001$) decrease compared to controls (*Fgfr1*$^{+/f}$::*Foxg1*$^{Cre/+}$) (*Figure 4—figure supplement 1A,B,E*), while the density of IHCs was not changed ($p > 0.09$). Furthermore, the length of *Fgfr1*$^{-/f}$::*Foxg1*$^{Cre/+}$ cochleae was only 9% shorter than control (*Figure 4—figure supplement 1A,B,F*), similar to what was observed in *Fgf20*$^{lacZ/lacZ}$ embryos (*Figures 1B, 2G* and ref. *Huh et al., 2012*). Whole mount Sox2 staining of E14.5 *Fgfr1*$^{-/f}$::*Foxg1*$^{Cre/+}$ cochleae was also comparable to control, indicating that the number of Sox2$^+$ progenitors was not changed (*Figure 4—figure supplement 1C*). In addition, proliferation of the

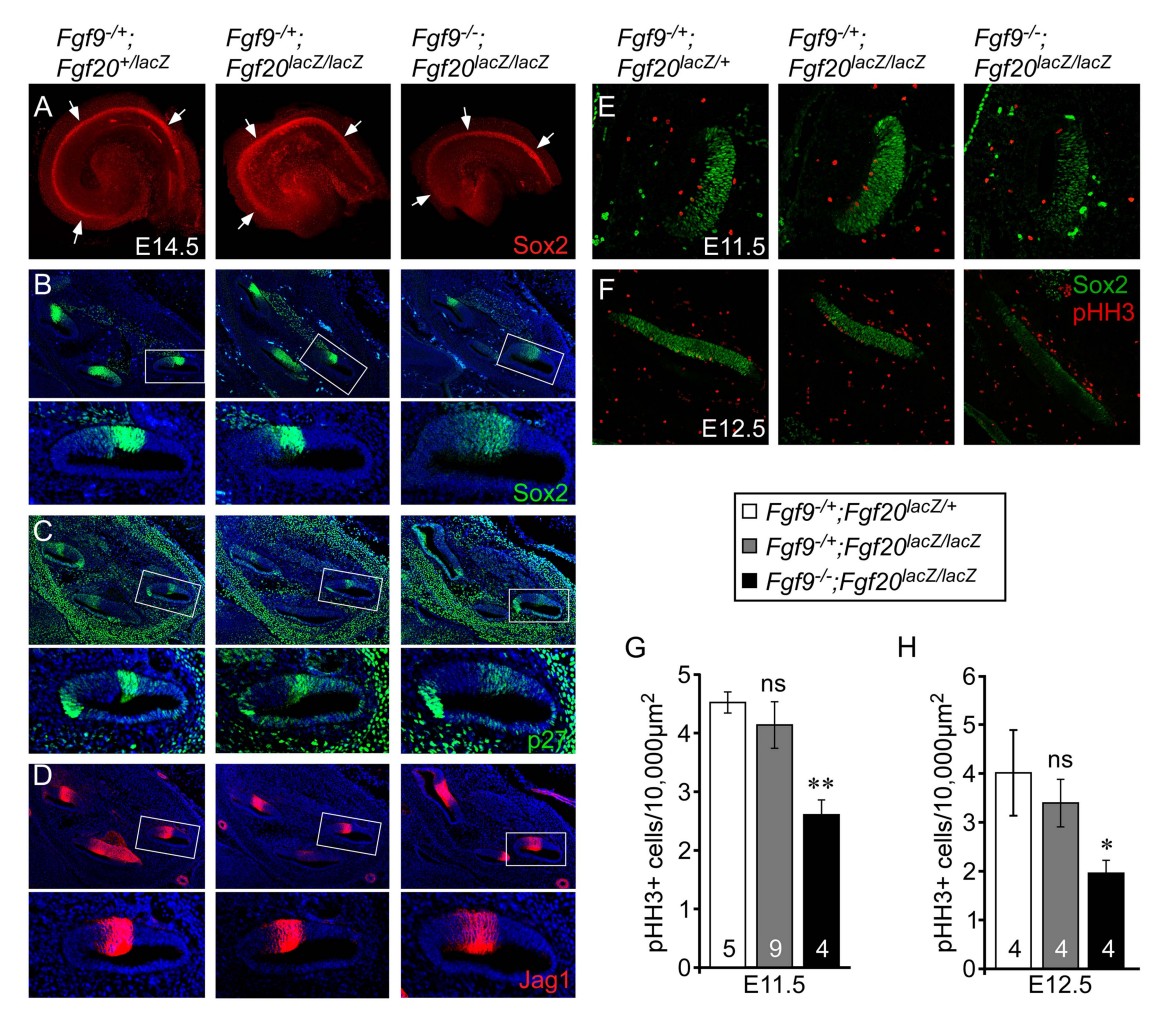

**Figure 3**. *Fgf9* and *Fgf20* are required for sensory progenitor proliferation. (**A**) Sox2 immunostaining of whole E14.5 cochlea to identify the progenitor domain (arrows). (**B–D**) Sox2 (**B**), p27 (**C**), and Jag1 (**D**) immunostaining of E14.5 *Fgf9⁻/⁺;Fgf20ˡᵃᶜᶻ/⁺*, *Fgf9⁻/⁺;Fgf20ˡᵃᶜᶻ/ˡᵃᶜᶻ*, and *Fgf9⁻/⁻;Fgf20ˡᵃᶜᶻ/ˡᵃᶜᶻ* embryo sections. Boxed regions of the cochlear duct are magnified below each image and were chosen in regions where the sections perpendicularly transect the cochlear duct. (**E**, **F**) Sox2 and phospho-Histone H3 (pHH3) co-immunostaining of E11.5 (**E**) and E12.5 (**F**) *Fgf9⁻/⁺;Fgf20ˡᵃᶜᶻ/⁺*, *Fgf9⁻/⁺;Fgf20ˡᵃᶜᶻ/ˡᵃᶜᶻ* and, *Fgf9⁻/⁻;Fgf20ˡᵃᶜᶻ/ˡᵃᶜᶻ* embryo sections. (**G**, **H**) Measurement of Sox2⁺ sensory progenitor proliferation at E11.5 (**G**) and E12.5 (**H**). All samples were compared with *Fgf9⁻/⁺;Fgf20ˡᵃᶜᶻ/⁺* double heterozygous controls. \*p < 0.01, \*\*p < 0.001. Sample numbers (n) are indicated in data bars. See also ***Figure 3—figure supplements 1, 2***.

The following figure supplements are available for figure 3:

**Figure supplement 1**. Proliferation of sensory progenitors.

**Figure supplement 2**. *Fgf9* and *Fgf20* loss do not cause premature cell cycle exit.

*Fgfr1⁻/ᶠ::Foxg1ᶜʳᵉ/⁺* prosensory epithelium was comparable (p > 0.6) to that of controls at E12.5 (***Figure 4—figure supplement 1D,G***).

Because *Fgfr2* often exhibits redundancy with *Fgfr1*, it is important to consider potential *Fgfr2* function in the inner ear prosensory epithelium. However, *Fgfr2* is required for formation of the otic vesicle and *Foxg1ᶜʳᵉ*, which is active before and during the otic vesicle stage (***Hébert and McConnell, 2000***), could not be used to investigate the role of *Fgfr2* at later stages of otic vesicle development. In addition, due to overall activity of *Foxg1ᶜʳᵉ* in the otic vesicle, cell type specificity of *Fgfr1* was still unknown. To study whether *Fgfr1* and/or *Fgfr2* function cell autonomously or non-cell autonomously

in the *Fgf20*[+] domain of the prosensory epithelium, we generated an *Fgf20*[Cre] allele (**Figure 4—figure supplement 2A**) to allow conditional gene targeting of the *Fgf20* lineage. To assess Cre activity, *Fgf20*[Cre/+];*ROSA*[mTmG/+] mice were generated. Cre activity was detected at E10.5 in a subset of the Sox2[+] prosensory domain, in a pattern identical to that of *Fgf20*[lacZ] embryos (**Figure 4—figure supplement 2B**). At P0, all of the components of the organ of Corti were positive for the *Fgf20*[Cre/+]; *ROSA*[mTmG/+] lineage tracer, indicating that *Fgf20*[Cre] is active in prosensory progenitors or their lineage (**Figure 4—figure supplement 2B**).

To identify potential roles for *Fgfr1* and *Fgfr2* in the prosensory epithelial lineage, we generated *Fgfr1* and *Fgfr2* single and double conditional mutant mice using the *Fgf20*[Cre] allele. E18.5 embryos were harvested and stained with phalloidin and p75, to visualize cochlear morphology. The phenotype of *Fgfr1*[−/f]::*Fgf20*[Cre/+](*Fgfr1*[−/f];*Fgfr2*[+/f]::*Fgf20*[Cre/+]) cochleae was similar to that of *Fgf20*[lacZ/lacZ], *Fgf9*[−/−];*Fgf20*[lacZ/lacZ], and *Fgfr1*[−/f]::*Foxg1*[Cre/+] cochleae (**Figures 2A–C, 4A–C, Figure 4—figure supplement 1A,B**). In contrast, the pattern and morphology of *Fgfr2*[−/f]::*Fgf20*[Cre/+](*Fgfr1*[+/f];*Fgfr2*[−/f]:: *Fgf20*[Cre/+]) cochleae was similar to control (**Figure 4A–C**) and *Fgfr1*[−/f];*Fgfr2*[−/f]::*Fgf20*[Cre/+] cochleae was comparable to *Fgfr1*[−/f]::*Fgf20*[Cre/+] (**Figure 4A–C**). The density of OHCs in *Fgfr1*[−/f];*Fgfr2*[−/f]::*Fgf20*[Cre/+] was comparable to *Fgfr1*[−/f]::*Fgf20*[Cre/+] (p > 0.94) and significantly (p < 0.001) decreased compared to control (**Figure 4F**). The density of OHCs of *Fgfr1*[−/f];*Fgfr2*[−/f]::*Fgf20*[Cre/+] embryos was comparable to control (p > 0.6) (**Figure 4F**) and the density of IHCs in *Fgfr1*[−/f]::*Fgf20*[Cre/+], *Fgfr2*[−/f]::*Fgf20*[Cre/+], and *Fgfr1*[−/f];*Fgfr2*[−/f]::*Fgf20*[Cre/+] cochleae were indistinguishable (ANOVA, p > 0.8) from that of controls (**Figure 4F**).

The length of the cochleae from E18.5 *Fgfr1*[−/f]::*Fgf20*[Cre/+] and *Fgfr1*[−/f];*Fgfr2*[−/f]::*Fgf20*[Cre/+] embryos was decreased by 19% and 25%, respectively, compared to controls (p < 0.0001, **Figure 4G**). However, the length of the cochleae from *Fgfr2*[−/f]::*Fgf20*[Cre/+] was comparable (p > 0.5) to controls. Together, these data, and those presented above, showed that epithelial *Fgfr1*, but not *Fgfr2*, is required for lateral compartment differentiation, and has a modest effect on cochlear duct length of a similar magnitude to the 10% reduction in cochlear length seen in *Fgf20*[lacZ/lacZ] mice (**Huh et al., 2012**). This reduction in cochlear length could be due to reduced numbers of progenitors or to other effects of FGFR1 signaling on cochlear duct elongation at later stages of development. Whole mount Sox2 staining at E14.5 of *Fgfr1*[−/f];*Fgfr2*[−/f]::*Fgf20*[Cre/+] cochleae showed a similarly sized sensory progenitor domain as compared to controls indicating that the Sox2[+] progenitor population was not affected by inactivation of epithelial *Fgfr1* and *Fgfr2* (**Figure 4D**). In addition, proliferation of *Fgfr1*[−/f]; *Fgfr2*[−/f]::*Fgf20*[Cre/+] cochleae was comparable (p > 0.5) to controls at E12.5 (**Figure 4E,H**).

## Mesenchymal *Fgfr1* and *Fgfr2* regulate cochlear length but not lateral compartment differentiation

We next asked whether mesenchymal FGFRs regulate cochlear length. *Twist2(Dermo1)*[Cre] is widely expressed in mesenchymal cells (**Li et al., 1995**; **Šošić et al., 2003**). To determine whether *Twist2*[Cre] is active in periotic mesenchyme during otic vesicle development, *Twist2*[Cre/+];*ROSA26*[lacZ/+] embryos were stained for lacZ activity at E9.5 and E10.5. lacZ activity was observed in all of the mesenchyme surrounding the unstained otic epithelium at both developmental time points (**Figure 5—figure supplement 1**). *Twist2*[Cre] was then used to inactivate *Fgfr1* and *Fgfr2* from mesenchymal cells. Cochleae were dissected from E18.5 embryos and stained with phalloidin and p75. All genotypes showed normal sensory HC and SC patterning, with one row of IHCs and three rows of OHCs (**Figure 5A–C**). The linear density of IHCs and OHCs was comparable (ANOVA, p > 0.2, p > 0.8, respectively) among all genotypes (**Figure 5F**). However, the length of the cochleae of *Fgfr1*[−/f]::*Twist2*[Cre/+] (*Fgfr1*[−/f];*Fgfr2*[+/f]::*Twist2*[Cre/+]), *Fgfr2*[−/f]::*Twist2*[Cre/+] (*Fgfr1*[+/f];*Fgfr2*[−/f]::*Twist2*[Cre/+]), and *Fgfr1*[−/f];*Fgfr2*[−/f]::*Twist2*[Cre/+] embryos were decreased by 7%, 20%, and 55%, respectively, compared to control (**Figure 5A,B,G**). Thus, the total number of HCs is decreased in proportion to the decreased length of the cochlea.

To determine whether the effect of loss of mesenchymal FGFRs on cochlear length originates early in development, we examined the size of the Sox2[+] progenitor domain at the time that HCs commit to differentiate, and cell proliferation within the Sox2[+] domain before the onset of differentiation. The size of the Sox2[+] progenitor domain, visualized by whole mount Sox2 staining of E14.5 cochleae was decreased in *Fgfr1*[−/f];*Fgfr2*[−/f]::*Twist2*[Cre/+] embryos compared to control embryos (**Figure 5D**). In addition, proliferation of Sox2[+] progenitors from *Fgfr1*[−/f];*Fgfr2*[−/f]::*Twist2*[Cre/+] cochleae was significantly (p < 0.01) decreased compared to control cochleae at E12.5 (**Figure 5E,H**). Together, these

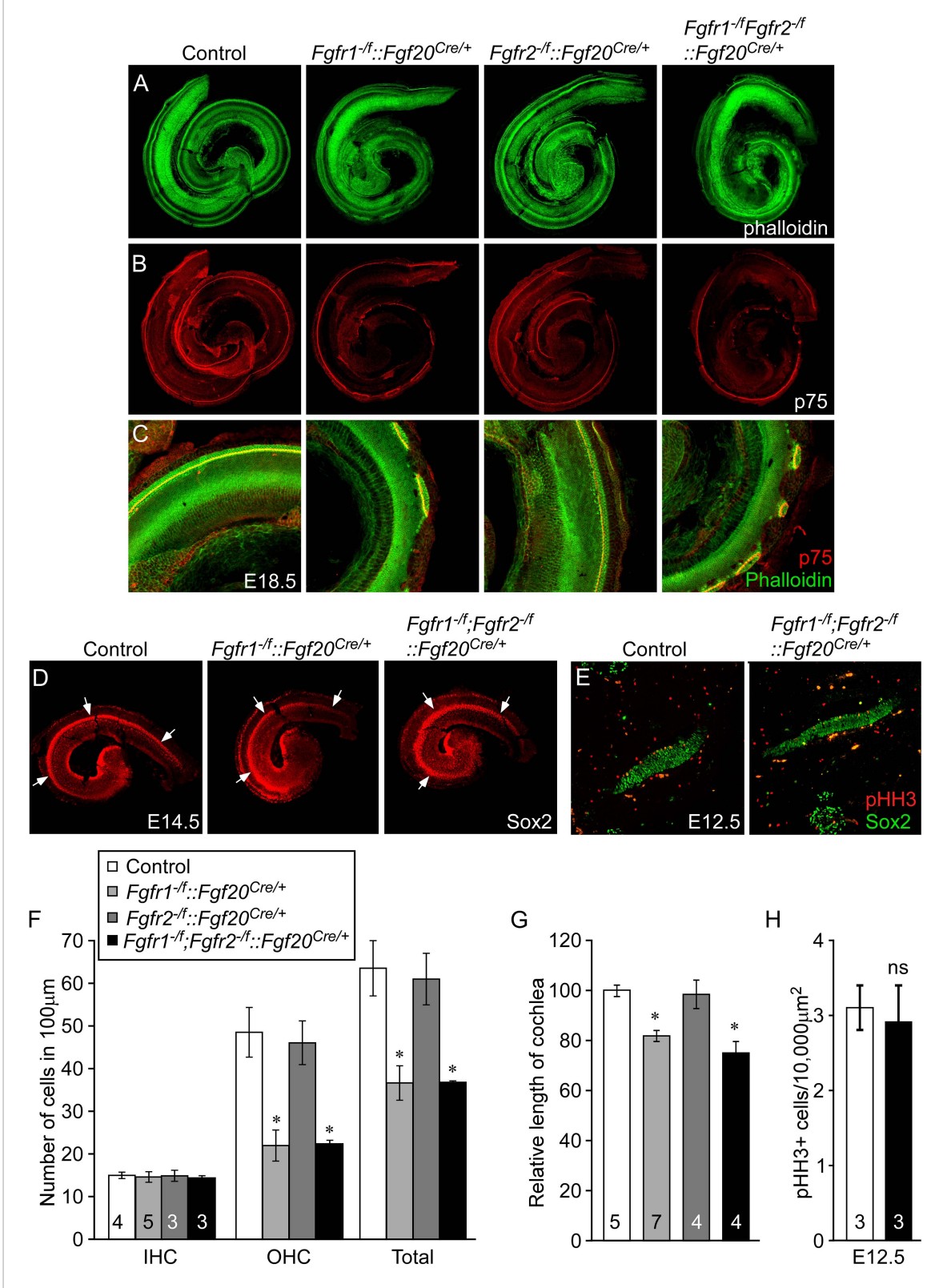

**Figure 4.** Cell-autonomous regulation of sensory progenitor differentiation requires epithelial *Fgfr1* but not *Fgfr2*. (**A**, **B**) Phalloidin (**A**) and p75 immunostaining (**B**) of E18.5 whole cochlea showing HCs (phalloidin) and pillar cells (p75) in the cochlear duct of control, *Fgfr1−/f::Fgf20Cre/+* (*Fgfr1−/f;Fgfr2+/f::Fgf20Cre/+*), *Fgfr2−/f::Fgf20Cre/+* (*Fgfr1+/f;Fgfr2−/f::Fgf20Cre/+*) and, *Fgfr1−/f;Fgfr2−/f::Fgf20Cre/+* embryos. (**C**) Phalloidin (green) and *Figure 4. continued on next page*

*Figure 4. Continued*

p75 immunostaining (red) showing the patterning of HCs and pillar cells in the cochlear duct. (**D**) Sox2 immunostaining of E14.5 whole cochlea to identify progenitor domains (arrows). (**E**) Sox2 and pHH3 co-immunostaining of E12.5 embryo sections. (**F**, **G**) Measurement of HC number (**F**) and length of cochleae (**G**) of E18.5 control embryos. (**H**) Measurement of Sox2+ sensory progenitor proliferation in E12.5 embryos. All samples were compared with controls. *p < 0.001; ns, not significant. Sample numbers (n) are indicated in data bars. See also *Figure 4—figure supplements 1, 2*.

The following figure supplements are available for figure 4:

**Figure supplement 1**. Epithelial *Fgfr1* is required for lateral compartment differentiation and HC and SC patterning.

**Figure supplement 2**. Generation of an *Fgf20^Cre^* knockin mouse line.

data show that mesenchymal FGFR signaling is a necessary determinant of cochlear length and sensory progenitor proliferation, but not for cochlear pattern formation or differentiation.

To determine whether the FGF signaling pathway is affected in periotic mesenchyme, whole mount RNA in situ hybridization was used to localize expression of *Etv4* and *Etv5*, two transcription factors that are commonly regulated by FGF signaling (*Raible and Brand, 2001*; *Firnberg and Neubüser, 2002*; *Brent and Tabin, 2004*; *Mao et al., 2009*; *Zhang et al., 2009*). Compared to double heterozygous control and *Fgf9^−/+^;Fgf20^lacZ/lacZ^* inner ears, *Fgf9^−/−^;Fgf20^lacZ/lacZ^* inner ears showed decreased expression of *Etv4* and *Etv5* in mesenchyme surrounding the cochlear duct (*Figure 6—figure supplement 1A,B*). The only known mesenchymal signaling pathway to regulate sensory progenitor proliferation is a Tbx1/Pou3f4 dependent retinoic acid (RA) signaling cascade (*Braunstein et al., 2008*, *2009*). However, *Tbx1* and *Pou3f4* expression, using RNA in situ hybridization in the embryos lacking *Fgf9* and *Fgf20* (*Fgf9^−/−^;Fgf20^lacZ/lacZ^*), did not reveal a change in expression of these transcription factors compared to doble heterozygous control and *Fgf9^−/+^;Fgf20^lacZ/lacZ^* embryos (*Figure 6—figure supplement 1C,D*), suggesting that FGF signaling may function independent of RA signaling.

## Mesenchymal FGF signal is sufficient to activate sensory progenitor cells proliferation

Next we asked whether increased mesenchymal FGFR signaling is sufficient to activate sensory progenitor proliferation. We ectopically expressed a constitutive FGFR1 tyrosine kinase domain in mesenchymal cells by combining the *Twist2^Cre^*, *ROSA^rtTA^*, and the doxycycline-responsive *TRE-caFgfr1-myc* alleles (*TRE-caFgfr1;ROSA^rtTA/+^::Twist2^Cre/+^*) (*Cilvik et al., 2013*). Doxycycline was fed to pregnant female mice beginning at E10.5. Embryos were analyzed at E12.5 for prosensory progenitor proliferation using pHH3 and Sox2 co-immunostaining (*Figure 6A*). The proliferation index in control prosensory cells was 2.4 ± 0.5/10,000 μm² (*Figure 6B*). However, embryos in which the *caFgfr1-myc* allele was induced in mesenchyme showed a significantly increased (4.1 ± 0.6/10,000 μm², p < 0.02) proliferation index (*Figure 6B*). To test the hypothesis that increased proliferation in sensory progenitors could lead to an increase in cochlear length, *TRE-caFgfr1;ROSA^rtTA/+^::Twist2^Cre/+^* embryos were induced from E10.5 to E14.5 and cochleae were analyzed at E18.5 (*Figure 6C*). Linear densities of IHCs, OHCs, and SCs in *TRE-caFgfr1;ROSA^rtTA/+^::Twist2^Cre/+^* embryos were comparable (p > 0.4) to control (*Figure 6E,F*). However, the cochlear length in *TRE-caFgfr1;ROSA^rtTA/+^::Twist2^Cre/+^* embryos was significantly (p < 0.001) increased by 14% compared to control (*Figure 6D*).

## Discussion

Sensory progenitor proliferation and differentiation are temporally distinct events in cochlear development. In mice, sensory progenitors exit from the cell cycle beginning at the apical end of the cochlea at ~E12.5 and ending at the base at ~E14.5. In contrast, differentiation begins in the mid-base at ~E14.5 and then extends to the base and apex (*Wu and Kelley, 2012*). Under physiological conditions, once progenitors exit the cell cycle, they do not reenter the cell cycle throughout the life of the organism. Previous studies suggested that during development both epithelial and mesenchymal signals are required to regulate cochlear progenitor proliferation and differentiation (*Montcouquiol and Kelley, 2003*; *Doetzlhofer et al., 2004*). However, the mechanisms that control cochlear sensory

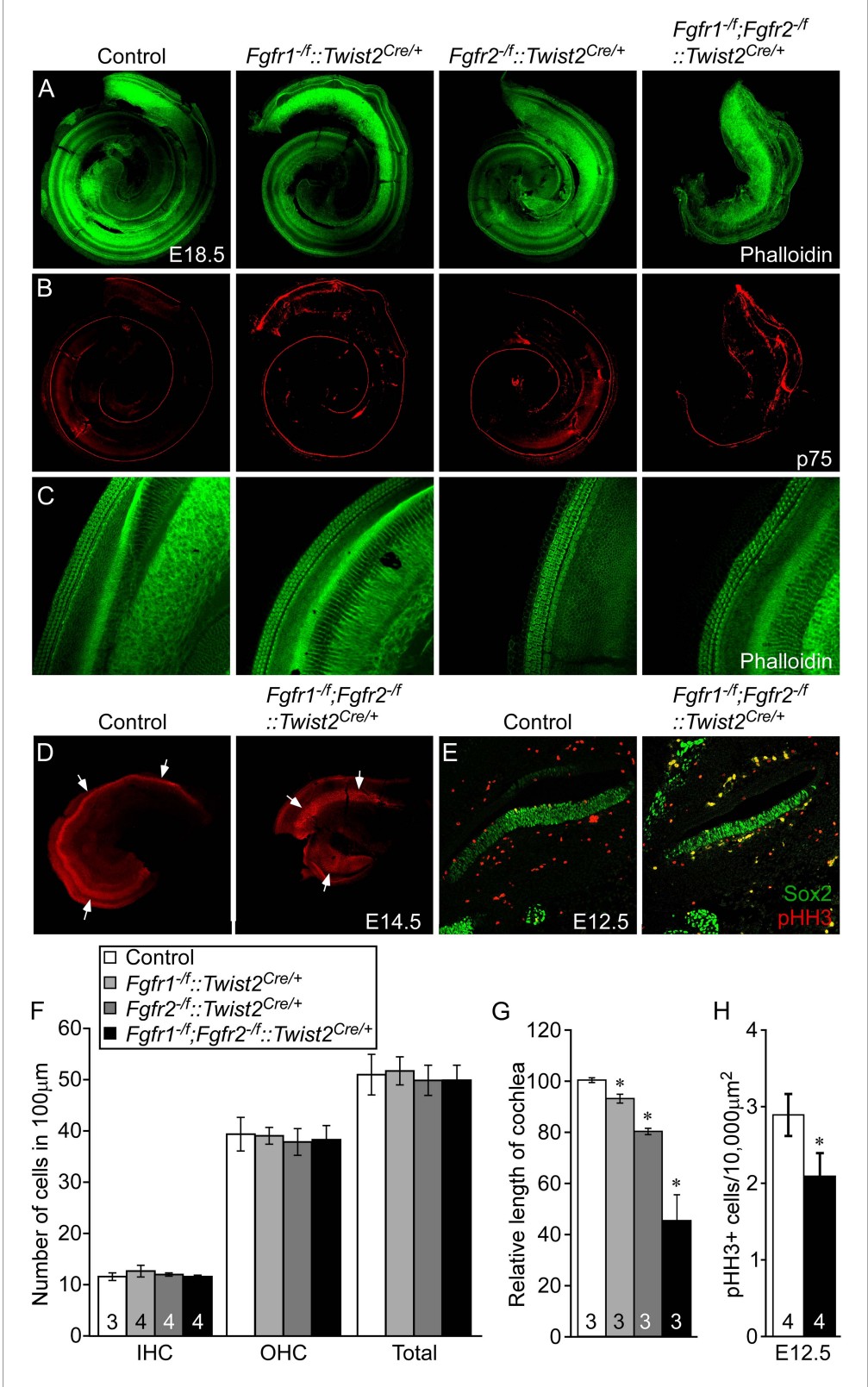

**Figure 5**. Mesenchymal *Fgfr1* and *Fgfr2* regulate the length of the cochlear duct and sensory progenitor proliferation. (**A**, **B**) Phalloidin (**A**) and p75 immunostaining (**B**) of E18.5 whole cochlea showing HCs (phalloidin) and pillar cells (p75) in the cochlear duct of control, *Fgfr1⁻/f::Twist2^Cre/+* (*Fgfr1⁻/f;Fgfr2+/f::Twist2^Cre/+*), *Fgfr2⁻/f::Twist2^Cre/+*

*Figure 5. continued on next page*

*Figure 5. Continued*

(*Fgfr1⁺/ᶠ;Fgfr2⁻/ᶠ::Twist2^{Cre/+}*), and *Fgfr1⁻/ᶠ;Fgfr2⁻/ᶠ::Twist2^{Cre/+}* embryos. (**C**) Phalloidin (green) staining showing normal HC patterning in the cochlear sensory epithelium. (**D**) Sox2 immunostaining of E14.5 whole cochlea to identify progenitor domains (arrows). (**E**) Sox2 and pHH3 co-immunostaining of E12.5 cochlea sections. (**F, G**) Measurement of HC number (**F**) and length of cochleae (**G**) of E18.5 control, *Fgfr1⁻/ᶠ::Twist2^{Cre/+}* (*Fgfr1⁻/ᶠ;Fgfr2⁺/ᶠ::Twist2^{Cre/+}*), *Fgfr2⁻/ᶠ::Twist2^{Cre/+}* (*Fgfr1⁺/ᶠ;Fgfr2⁻/ᶠ::Twist2^{Cre/+}*), and *Fgfr1⁻/ᶠ;Fgfr2⁻/ᶠ::Twist2^{Cre/+}* embryos. (**H**) Measurement of Sox2⁺ sensory progenitor (green) proliferation (red, pHH3) in E12.5 embryo sections. All samples were compared with controls. *p < 0.001. Sample numbers (n) are indicated in data bars. See also ***Figure 5—figure supplement 1***.
The following figure supplement is available for figure 5:

**Figure supplement 1**. *Twist2^{Cre}* targeting of periotic mesenchyme.

progenitor proliferation are not known. In this study, we found that epithelial FGF9 and FGF20 signaling to mesenchymal FGFR1 and FGFR2 is required for normal levels of cochlear sensory progenitor proliferation and that inactivation of either the ligands or the mesenchymal receptors results in a shortened cochlea. We also demonstrated that activation of mesenchymal FGFR signaling is sufficient to increase sensory progenitor proliferation and extend cochlear length.

*Fgf9* is expressed in non-sensory epithelia of the cochlea and loss of *Fgf9* results in defects in periotic mesenchymal cell proliferation, causing a hypoplastic otic capsule (***Pirvola et al., 2004***). Based on known expression patterns in mesenchyme, *Fgfr1* and *Fgfr2* were considered the most likely targets of FGF9 signaling (***Pirvola et al., 2004***). The critical time window for FGF9 signaling was determined to occur before E14.5. By contrast, *Fgf20* is expressed in the sensory epithelium and loss of *Fgf20* results in failure of the lateral compartment of the organ of Corti to fully differentiate (***Huh et al., 2012***). Based on expression patterns and phenotypic similarities with epithelial *Fgfr1* conditional gene inactivation, FGFR1 was identified as the epithelial target receptor (***Pirvola et al., 2002***; ***Huh et al., 2012***).

The effects of epithelial FGFR1 signaling on the length of the cochlear duct exhibit variability among studies. ***Ono et al. (2014)*** report a 40–50% decrease in cochlear length in *Fgfr1^{f/f}::Six1enh21^{Cre}*, and *Fgfr1^{f/f}::Emx2^{Cre}* conditional knockout mice, by contrast, *Fgf20^{lacZ/lacZ}*, *Fgf9⁻/⁺;Fgf20^{lacZ/lacZ}*, and *Fgfr1⁻/ᶠ::Foxg1^{Cre/+}* mice that we studied (***Huh et al., 2012***, and this study) showed only a 10–25% decrease in cochlear length. It is clear that in both studies defects in epithelial differentiation is likely to result in some decrease in cochlear length. It is also possible that differences in genetic background could contribute to differences in these two studies.

FGF9 and FGF20 are members of the same FGF subfamily and share similar biochemical properties (***Zhang et al., 2006***; ***Ornitz and Itoh, 2015***). Redundancy between these FGFs has also been demonstrated in kidney development, where both ligands are required for nephron progenitor maintenance (***Barak et al., 2012***). Interestingly, in both cases, the expression patterns of these two FGFs do not overlap, but nevertheless they appear to signal to a common target tissue, periotic mesenchyme in the developing inner ear and CAP mesenchyme in the developing kidney. For the evolution of the kidney and inner ear, it is possible that additive expression of these FGFs from distinct sources was required to take advantage of their unique receptor specificities or unique interactions with the extracellular matrix.

Tbx1 is a transcription factor that is expressed in both sensory epithelium and mesenchyme (***Vitelli et al., 2003***; ***Raft et al., 2004***). Deletion of *Tbx1* in mesenchymal cells resulted in defects in cochlear epithelial proliferation indicating a non-cell autonomous requirement for Tbx1 for cochlear epithelial development (***Xu et al., 2007***). In addition, the Pou domain containing transcription factor, Pou3f4, also known as Brn4, is expressed in mesenchymal cells in the developing inner ear (***Phippard et al., 1999***). Deletion of *Pou3f4* resulted in reduction of cochlear length and defects in derivatives of the otic mesenchyme including the spiral limbus, scala tympani, and strial fibrocytes (***Phippard et al., 1999***). Furthermore, decreasing gene dosages of *Tbx1* and *Pou3f4* resulted in a significant decrease in sensory epithelial proliferation and cochlear length indicating that Tbx1 and Pou3f4 genetically interact. The RA catabolizing genes *Cyp26a1* and *Cyp26c1*, both targets of Tbx1 and Pou3f4, were decreased in these mice, suggesting that increased RA signaling could directly or indirectly suppress sensory progenitor proliferation (***Braunstein et al., 2008***, ***2009***). Analysis of *Fgf9* and *Fgf20* double

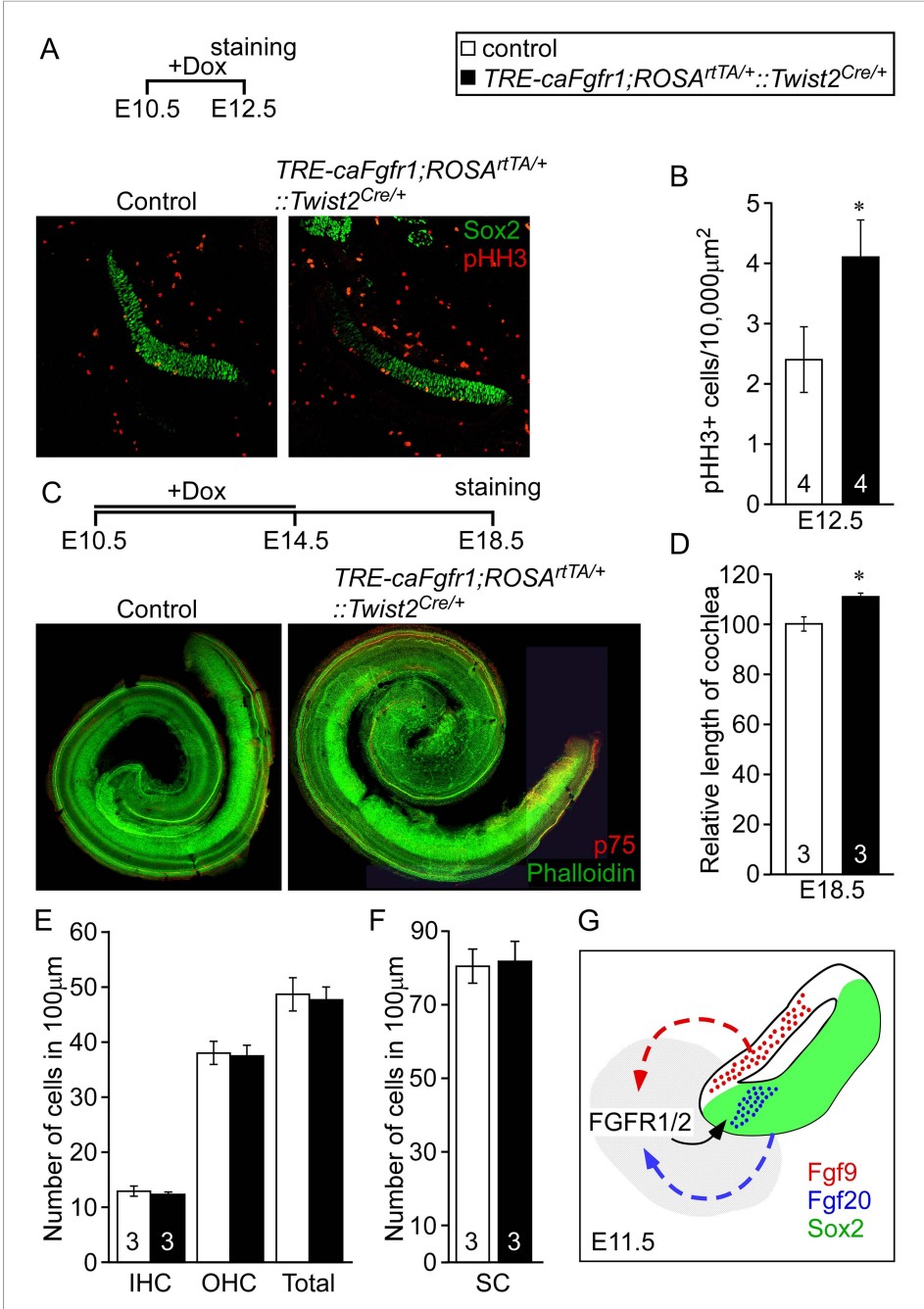

**Figure 6**. Ectopic activation of FGFR signaling in mesenchyme increases sensory progenitor proliferation and cochlear length. (**A**) Sox2 (green) and pHH3 (red) co-immunostaining of E12.5 control and *TRE-caFgfr1;ROSA^rtTA/+::Twist2^Cre/+* embryo sections. (**B**) Measurement of Sox2+ sensory progenitor proliferation at E12.5. (**C**) Phalloidin (green) and p75 immunostaining (red) showing the patterning of HCs and pillar cells in the cochlear duct of control and *TRE-caFgfr1;ROSA^rtTA/+::Twist2^Cre/+* embryos. Measurement of the length of the cochleae (**D**), HC number (**E**), and SC number (**F**) of E18.5 control and *TRE-caFgfr1;ROSA^rtTA/+::Twist2^Cre/+* embryos. (**G**) Schematic diagram indicating the requirement for epithelial FGF9/20 signaling to mesenchymal FGFR1/2 to induce sensory progenitor proliferation. *p < 0.01 in **B** and *p < 0.001 in **D**. Sample numbers (n) are indicated in data bars. See also *Figure 6—figure supplement 1*.

The following figure supplement is available for figure 6:

**Figure supplement 1**. *Fgf9* and *Fgf20* regulate the expression of *Etv4* and *Etv5*, but not *Pou3f4* or *Tbx1*.

mutant mice showed no change in the expression of Tbx1 and Pou3f4 in mesenchyme surrounding the otic vesicle, suggesting that mesenchymal FGF signaling does not directly affect transcription factors that regulate RA signaling (*Figure 6—figure supplement 1C,D*). On the other hand, *Etv4* and *Etv5* function as downstream targets of FGF signaling in other systems including the limb (*Mao et al., 2009*; *Zhang et al., 2009*), and *Etv4* and *Etv5* expression were decreased in *Fgf9/Fgf20* mutant ears. Future studies will be needed to determine whether FGF signaling including ETV4 and ETV5 regulates RA signaling downstream of Tbx1/Pou3f4 or act in parallel to the Tbx1/Pou3f4/RA signaling pathway to regulate sensory progenitor proliferation. Whether the cellular target of RA signaling is in the periotic mesenchyme or the sensory progenitor epithelium also remains to be determined. It is also possible that the number of nearby mesenchymal cells may influence sensory progenitor proliferation. However, considering that loss of *Fgf9* resulted in decreased mesenchymal cell proliferation (*Pirvola et al., 2004*) but did not affect HC formation or cochlear length (*Figure 2*), alternative mechanisms may need to be considered.

The reactivation of developmental signaling pathways may be important for regeneration. Recent publications showed that inhibition of Notch signaling could induce transdifferentiation of SCs to HCs in a damaged cochlea (*Korrapati et al., 2013*; *Mizutari et al., 2013*). In addition, Wnt/β-catenin signaling can induce SC proliferation in neonatal mice (*Chai et al., 2012*; *Shi et al., 2012*). One intermediate goal of regenerative biology for the inner ear would be to generate large numbers of sensory progenitor cells that could be differentiated into functional HCs and SCs and then be reintroduced into the damaged inner ear. The studies presented here suggest that in efforts to grow inner ear sensory progenitor cells in vitro, that FGF-induced mesenchyme may be necessary. The identification of mesenchymal factors that are regulated by FGF or RA could also be used to support the growth of sensory progenitor cells.

## Materials and methods

### Generation of *Fgf20^Cre^* mutant mice

This study was carried out in strict accordance with the recommendations in the Guide for the Care and Use of Laboratory Animals of the National Institutes of Health. The protocol was approved by the Washington University Division of Comparative Medicine Animal Studies Committee (Protocol Number 20130201). All efforts were made to minimize animal suffering.

*Fgf20^Cre^* knock-in mice were generated using a similar method to that reported previously (*Huh et al., 2012*). Briefly, exon1 of *Fgf20* was replaced with a Cre-EGFP–FRT-neomycin-FRT cassette to generate *Fgf20^Cre(neo)/+^* mice. The neomycin gene was eliminated by mating with *CAG-FLPe* (*Kanki et al., 2006*) mice to generate *Fgf20^Cre/+^* mice. Genotyping was performed using PCR1: CTGCATTC GCCTCGCCA CCCTTGCTACACT; PCR2: GGATCTGCAGGTGGAAGCCGGTGCGGCAGT; PCR3: TTCAGGGT CAGCTTGCCGTAGGTGGCATCG primers, which amplify wild type (335 bp) and mutant (241 bp) PCR fragments. Mice were maintained on a 129X1/SvJ;C57BL/6J mixed background.

### Generation of *Fgf9^lacZ^* mutant mice

*Fgf9^lacZ^* mice were derived from International Knockout Mouse Consortium targeted ES cells (project number 24486) (*Skarnes et al., 2011*). Chimeric mice derived from injected blastocysts were bred to *Sox2^Cre^* mice (*Hayashi et al., 2003*) to remove the nbactP-neo selection cassette and the second exon of the *Fgf9* gene. Genotyping was performed using Wt1: GAAGTCGTGCGTGAGGTGCTCCAGG TCGG; Wt2: CCGCGAATGCTGACCAGGCCCACTGCTAT primers for wild type (172 bp) and mut1: GTT GCA GTGCACGGCAGATACACTTGCTGA; mut2: GCCACTGGTGTGGGCCATAATTCAATTCGC primers for mutant (389 bp) PCR fragments. Mice were maintained on a 129X1/SvJ;C57BL/6J mixed background.

### Other mouse lines

*Fgfr1^f/f^*, *Fgfr2^f/f^*, *Twist2(Dermo1)^Cre/+^*, *Foxg1^Cre/+^*, *R26R*, *ROSA^mTmG/+^*, *TRE-caFgfr1-myc*, *ROSA^rtTA/+^*, *Fgf20^lacZ/+^*, and *Fgf9^−/+^* mice lines were reported previously (*Soriano, 1999*; *Hébert and McConnell, 2000*; *Colvin et al., 2001*; *Pirvola et al., 2002*; *Šošić et al., 2003*; *Yu et al., 2003*; *Belteki et al., 2005*; *Muzumdar et al., 2007*; *Huh et al., 2012*; *Cilvik et al., 2013*). *Fgfr1^−/+^* and *Fgfr2^−/+^* mice were generated by crossing *Fgfr1^f/f^* and *Fgfr2^f/f^* to *Sox2^Cre/+^* mice, respectively.

## βGal staining

Embryos were fixed overnight in Mirsky's Fixative (National Diagnostics, Atlanta, GA), washed three times in PBT (PBS, 0.1% Tween-20) and incubated in βGal staining solution (2 mM $MgCl_2$, 35 mM potassium ferrocyanide, 35 mM potassium ferricyanide, 1 mg/mg X-Gal in PBT) at 37°C until color reaction was apparent. Samples were washed in PBS, fixed in 10% formalin and imaged under a dissecting microscope.

## Histology

For frozen sections, embryos were fixed with 4% paraformaldehyde overnight and washed with PBS. Samples were soaked in 30% sucrose and embedded in OCT compound (Tissue-Tek, Torrance, CA). Samples were sectioned (12 µm) and stored at −80°C for immunohistochemistry.

## Quantification of HC and SC numbers

Either phalloidin or Prox1 immunostaining were used to identify HCs and SCs, respectively. To measure the density of HCs and SCs, at least 300 µm regions of the base (10%), middle (40%), and apex (70%) of the cochleae were counted and normalized to 100 µm along the length of the cochlear duct. Inner and OHCs were identified by location and morphology of phalloidin staining. Cell counting was performed using Image J software.

## Proliferation and cell death analyses

To analyze progenitor proliferation and cell death, frozen sections were prepared from the entire ventral inner ear of E11.5 or E12.5 embryos. Alternate sections were subjected to staining for pHH3 and Sox2 (for proliferation) or activated-Caspase 3 and Sox2 (for cell death, data not shown). For EdU labeling, pregnant females were injected with 50 µg/g (body weight) of EdU according to the manufacture's recommendation. Embryos were collected 2 hr after EdU injection. EdU was detected with the Click-iT EdU Alexa Fluor 488 Imaging Kit (Invitrogen, Carlsbad, CA) according to manufacture's instructions. The total area of $Sox^+$ cells was measured using Image J software and $pHH3^+$ or activated-$Caspase\ 3^+$ cells within the $Sox2^+$ domain were counted. Counting was normalized to 10,000 $µm^2$ of $Sox2^+$ prosensory epithelium.

## Immunohistochemistry

For whole mount immunofluorescence, cochleae were isolated and fixed in 4% PFA overnight at 4°C. Samples were washed with PBS and blocked with PBS containing 0.1% triton X-100 and 0.5% donkey serum. Primary antibody was incubated overnight at 4°C. Samples were washed with PBS and incubated with a secondary antibody for 1 hr at room temperature. Samples were washed, placed on a glass microscope slide, coverslipped, and photographed using a Zeiss LSM 700 confocal microscope. For immunofluorescence on histological sections, frozen sections (12 µm) were washed with PBS, blocked with 0.1% triton X-100 and 0.5% donkey serum, and incubated with primary antibodies in a humidified chamber overnight at 4°C. Sections were then washed and incubated with secondary antibody for 1 hr at room temperature. Samples were washed, coverslipped with Vectashield Mounting Media (Vector Labs, Burlingame, CA), and photographed using a Zeiss LSM 700 confocal microscope. Primary antibodies used: Phallodin (R&D Systems, Minneapolis, MN, 1:40), Prox1 (Covance, Princeton, NJ, 1:250), p27 (Neomarkers, Fremont, CA, 1:100), p75 (Chemicon, Billerica, MA, 1:500), β-galactosidase (Abcam, United Kingdom, 1:500), Sox2 (Millipore, Billerica, MA, 1:250, Santa Cruz, Dallas, TX, 1:250), Jag1 (Santa Cruz, Dallas, TX, 1:200), phospho-histone 3 (Sigma–Aldrich, St. Louis, MO, 1:500), and activated Caspase 3 (BD Sciences, San Jose, CA, 1:200).

## Statistics

Numbers of samples are indicated for each experiment. All data are presented as mean ± standard deviation (sd). The p value for difference between two samples was calculated using a two-tailed Student's $t$-test or one-way ANOVA where appropriate. $p < 0.05$ was considered as significant.

## Acknowledgements

We thank Craig Smith for technical help. This work was funded with a grant from the Action on Hearing Loss Foundation (DMO), the Office of Naval Research N000141211025 (MEW), the March of

Dimes Foundation (DMO), the Hearing Health Foundation (SHH), NIH K99 DC012825 (SHH). Confocal microscopy was supported by the Microscopy & Didigal Imaging Core (NIH P30 DC004665). Mouse lines were generated with assistance from the Mouse Genetics Core, the DDRCC Murine Models Core Grant (NIH P30 DK052574), and the Washington University Musculoskeletal Research Center (NIH P30 AR057235).

## Additional information

### Funding

| Funder | Grant reference | Author |
| --- | --- | --- |
| Action on Hearing Loss | | David M Ornitz |
| March of Dimes Foundation (March of Dimes Births Defect Foundation) | | David M Ornitz |
| National Institutes of Health (NIH) | K99 DC012825 | Sung-Ho Huh |
| Hearing Health Foundation (HHF) | | Sung-Ho Huh |
| Office of Naval Research (ONR) | N000141211025 | Mark E Warchol |
| National Institutes of Health (NIH) | Microscopy & Digital Imaging Core P30 DC004665 | Mark E Warchol |
| National Institutes of Health (NIH) | DDRCC Murine Models Core Grant P30 DK052574 | David M Ornitz |
| National Institutes of Health (NIH) | Washington University Musculoskeletal Research Center P30 AR057235 | David M Ornitz |

The funders had no role in study design, data collection and interpretation, or the decision to submit the work for publication.

### Author contributions

SHH, Conception and design, Acquisition of data, Analysis and interpretation of data, Drafting or revising the article; MEW, Analysis and interpretation of data, Drafting or revising the article; DMO, Conception and design, Analysis and interpretation of data, Drafting or revising the article

### Author ORCIDs

David M Ornitz, http://orcid.org/0000-0003-1592-7629

### Ethics

Animal experimentation: This study was carried out in strict accordance with the recommendations in the Guide for the Careand Use of Laboratory Animals of the National Institutes of Health. The protocol was approved by the Washington University Division of Comparative Medicine Animal Studies Committee (Protocol Number20130201). All efforts were made to minimize animal suffering.

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
