## [Decision Letter]

Thank you for sending your work entitled “Cochlear progenitor number is controlled through mesenchymal FGF receptor signaling” for consideration at *eLife*. Your article has been evaluated by Janet Rossant (Senior editor) and three reviewers, one of whom is a member of our Board of Reviewing Editors.

The following individuals responsible for the peer review of your submission have agreed to reveal their identity: Tanya Whitfield (Reviewing editor); Raj Ladher (peer reviewer). A further reviewer remains anonymous.

The Reviewing editor and the other reviewers discussed their comments before we reached this decision, and the Reviewing editor has assembled the following comments to help you prepare a revised submission.

As you will see, all three reviewers found the work interesting and potentially important. However, they felt that the data, as they stand, do not provide full support for the proposed model, and that further work and clarification is required. As a minimum, this should include a full characterisation of the mesenchymal phenotype in each of the mutants, including the *Fgf9* and *Fgf20* single mutants for comparison. Improved quantitation of the sensory cell phenotype and levels of proliferation in epithelium and mesenchyme, ideally including a marker other than PHH3, will be important. Alternative interpretations of the mesenchymal phenotype should be considered and discussed. A fuller discussion of the phenotype caused by *Fgfr1* KO in the epithelium is also required, since this also affects cochlear length. The revised version must include citation, acknowledgement and discussion of previously published work that impacts on the current study. In particular, it will be worth examining the IHC phenotype carefully here.

The full reviews are appended below.

Reviewer #1:

This is an interesting paper that builds on a previous study (18) demonstrating a role for *Fgf2*0 in differentiation of the lateral compartment of the cochlea. The new information here is that FGF signalling from epithelium to mesenchyme is both required and sufficient for sensory progenitor proliferation and extension of the cochlear duct. The data are clear and support the conclusions well. However, there is no identification of a potential mechanism for the mesenchymal signal back to the cochlear duct that regulates its growth. As a result, the advance that the study makes over previous work appears to be somewhat incremental.

A few details of the manuscript could be clarified:

The normal expression patterns of *Fgfr*1 and *Fgfr2* in the otic epithelium and periotic mesenchyme at the relevant stages should be presented or referred to early on in the manuscript.

Introduction: ‘…and that mesenchymal FGF signaling is sufficient to promote…’. This is ambiguous. Is this the same signalling from epithelium to mesenchyme as referred to in the previous sentence, or new signalling from mesenchyme to epithelium, also mediated by FGF?

Figure 1 and Results: Make sure the text follows the order of the panels in the figure and vice versa—i.e. show *Fgf20* first in the figure.

Figure 1: The sketch here does not appear to represent the expression patterns shown very well. The two FGF domains appear to abut each other directly, with a very sharp boundary at the junction between non-sensory (thinner) and sensory (thicker) epithelium. This is not shown in the sketch, which does not depict the full extent of the *Fgf20* domain and shows a substantial gap between the two domains.

Figure 2: It would be helpful to show the *Fgf20*^*-/-*^; *Fgf9*^*+/+*^ phenotype here, to illustrate the effects of the loss of *Fgf20* alone, even though previously published.

In the subsection headed “*Fgf9* and *Fgf20* regulate cochlear length and *Fgf20*, not *Fgf9*, is required for lateral compartment differentiation and patterning”, when you state “…the normal pattern of three rows of OHCs… (Figure 2)”, there should be a reference to the later panels (C, D) here as well.

Figure 2: Please mark on the positions of IHC and OHC on these panels. It looks to me as if there are two rows of IHC in the *Fgf20*^*βGal/βGal*^, especially in the right hand panel. Please comment.

The density of each HC type is measured for several different genotypes. It should be clarified somewhere that where patterning of the organ of Corti is affected, the whole organ is fragmented and that cell measurements are (presumably) taken over patches where the sensory cells are still present. If density was measured along the whole length of the cochlea, presumably IHC density would also be affected.

Figure 3: Other areas marked by Jag1, but not highlighted, appear to show significant differences from controls, e.g. in the top left hand corner of the panel, the domain appears to be twice as big as in controls, whereas the stained area towards the bottom of the panel appears much reduced. Please comment.

The Discussion is rather brief, and does not really exploit the detail provided in the Results. In particular, the Discussion does not cover the potential differences in roles between *Fgf9* and *Fgf20*, and *Fgfr1* and *Fgfr2*. My interpretation of the Results was that *Fgf20* primarily signals through *Fgfr1* in the cochlear epithelium to regulate patterning of lateral compartment of the organ of Corti, while *Fgf9* primarily signals through *Fgfr2* in the mesenchyme to regulate cochlear length, with some cross-activity and redundancy between the two FGF pathways. However, the Discussion merely focusses on the redundancy rather than any differences between the two. It would be interesting to relate any differences to the clear differences between the non-overlapping expression domains of each FGF in the developing cochlear duct, and to the expression domains of the FGFRs, which are not shown (see comment above).

In addition, there is no mention of how the mesenchyme may signal back to the epithelium to regulate cochlear growth. It is not made clear in the paragraph about RA signalling whether this is being proposed as the mechanism of mesenchymal to epithelial signalling, but the last sentence of the manuscript appears to indicate that this may be a second signalling pathway that is regulating mesenchymal cell behaviour. Do the authors have any speculation on how the mesenchyme may regulate cochlear duct growth? Are there any candidate factors?

Reviewer #2:

*Fgfr1* signalling has been implicated in the regulation of the pool of sensory progenitors that give rise to the hair cells and support cells of the mammalian cochlea and in the specification of outer hair cells during later organ of Corti patterning. In previous studies, Huh and colleagues described the role of *Fgf20* in the mediating the latter function. In the current study the authors analyse the role of *Fgf9* and *Fgf20* in the control of the sensory progenitor pool. Using Cre-drivers they suggest that the ligands signal via the mesenchyme and back on to the otic epitehlia.

One of my major criticisms of this paper is the failure to reconcile their data with data that is already published. A recent study of the *Fgfr1* phenotype (33) has not been considered and needs to be evaluated in the context of the authors work. For example, in the subsection “Epithelial *Fgfr1* but not *Fgfr2* is required for lateral compartment differentiation”, the sentence “however, no quantitative data regarding affected cell types or cochlear length was reported” (in *Foxg1*^*Cre*^
*Fgfr1* mutants) is untrue—the data is presented in Ono et al. Similarly, a paper showing the role of *Fgf20* in the maintenance of Sox2^+^ progenitors (Munnamalai et al., 2012) has not been cited or considered.

With this in mind, it is probably worth highlighting the points that are inconsistent:

a) In *Fgf9/20* knockouts, *Fgf20*^*Cre*^;*Fgfr1/2* and *Dermo1*^*Cre*^;*Fgfr1/2* mutants the inner hair cells are unaffected. In Ono et al., early deletion (using Six1-Cre and *Foxg1*^*Cre*^) of *Fgfr1* affects IHC; Cochlear length is affected, quite significantly, in the early, epithelial deletions of *Fgfr1* presented by Ono et al. Huh, Warchol and Ornitz do not find any reduction in epithelial size; Proliferation of Sox2^+^ progenitors was unaffected in Ono's early deletions of *Fgfr1*.

Alternative interpretations of the mesenchymal requirement of FGFR signalling in the regulation of cochlear length need to be considered. One possibility is that the effect on cochlear length is secondary, not because of a direct action of *Fgf9/20* on *Fgfr1* and *2* in the mesenchyme, but because there is simply less mesenchyme. I think that it is quite likely that a reduction of *Fgfr1* and *2* in the peri-otic mesenchyme at such early stages leads to a reduction in the amount of mesenchyme. As Doetzlhofer et al., had shown, the mesenchyme is important for outgrowth. The authors need to analyse the periotic mesenchyme in these mutants.

The correlation of sensory progenitors with cochlea length is quite a laboured point. It is clear that even in the absence of Sox2^+^ progenitors (see [22]), the cochlea still extends. It is likely that progenitor proliferation plays at best a contributory role.

The cochlear of *Fgf9* nulls, as well as the *Fgf20* nulls and *Fgf9/20* double nulls should also be analysed. Are these the same as wild-types or is the *Fgf9* and *Fgf20* phenotype additive?

Reviewer #3:

This is an interesting study that attempts to show the relative importance of *Fgf9* and *Fgf20* on the development of the prosensory domain of the mouse cochlea. The control of cell number in the prosensory domain of the cochlea is an important and unresolved issue in the field, as there is no apparent cell death, and so the number of progenitors generated directly affects the size of the organ of Corti.

In the first part of the paper, the authors show a role for *Fgf9* on cochlear length; while *Fgf20* seems to be affecting the number of prosensory progenitors that are being generated, or the way they are being patterned. This role for *Fgf20* has been previously described by this group, and here they extend that work to show that *Fgf9* does not play a role in this aspect. In Figure 2, the paper suffers from not showing data from *Fgf20*^*-/-*^ mice alone (presumably because this is contained in their earlier publication). I think evidence from this mouse should be included here as part of the current analysis, so that direct comparison is possible; also, its absence makes the description of the various phenotypes rather confusing, and the authors might want to spend some time thinking of a way to describe this part more clearly. In the end of this first section, the authors claim that there are synergistic effects between the two genes; but I am unclear about why they are making this interpretation, since the lack of any patterning defects in *Fgf9*^*-/-*^, seems to suggest that this FGF acts outside the prosensory domain to affect the length of the cochlear duct (Figure 2).

The main point of the second section (Figure 3), is that *Fgf9* plays a synergistic role with *Fgf20* in the proliferation of the cells of the prosensory domain that will make up the organ of Corti. This is an interesting observation, but seems to contradict the separate roles that the two genes play as described in Figure 2, namely prosensory (*Fgf20*) and non-prosensory (*Fgf9*). The data here are also a little difficult. In Figure 3, they are counting cells in histological sections rather than whole mounts? But it is difficult to know how much of the prosensory domain they actually measure, and did they count in exactly the same basal-apical position? This may have been difficult to do, since the (*Fgf20*^*-/-*^*::Fgf9*^*-/+*^) does show a change in patterning (“less compact”). Thus, the modest counting difference that they observe in Figure 3 could be due to differences in cell distribution within the duct? Indeed, from this figure there does seem to be a trend in their data showing that *Fgf20* has an effect on its own, without *Fgf9*, even if their statistical test does not show a significant difference; and so *Fgf9* could be skewing the data by changing the shape of the duct? This leads me to be skeptical of their interpretation that the genes are acting together, instead of independently on two different elements of the observed phenotype.

Figure 5: I don't think the conclusion that “together, these data show that FGFR1/2, expressed in mesenchymal cells, are required for proper cochlear length formation and for sensory progenitor proliferation, but not for cochlear pattern formation or differentiation” is completely supported in the case of proliferation, since they fail to isolate the two genes for the proliferation part of the study (Figure 5), and only look at the double KO. How do we know that FGFR1 is not responsible, on its own, for the proliferation defect, an observation that would be consistent with the previous Pirvola study, I think?

Figure 6: In which they show that activation of FGFR signalling is sufficient to affect prosensory proliferation, is very interesting, and complements the KO data.

Discussion:

The authors state that their evidence shows that “in this study, we found that epithelial FGF9 and FGF20 signaling to mesenchymal FGFR1 and FGFR2 is required for normal levels of cochlear sensory progenitor proliferation and that inactivation of either the ligands or the mesenchymal receptors results in a shortened cochlea.”

Also, that “…the expression patterns of these two FGFs do not overlap, but nevertheless they appear to signal to a common target tissue, periotic mesenchyme in the developing inner ear…”

But how do we know that pro-sensory-derived FGF9 and FGF20 are signaling through mesenchymal FGFR1/2? Why couldn't they also be signaling through epithelial FGFRs? The fact that overstimulation of FGFR1/2 in mesenchyme changes the epithelial proliferation may suggest this is one route, but does not prove this, I don't think.

[Editors' note: further revisions were requested prior to acceptance, as described below.]

Thank you for resubmitting your work entitled “Cochlear progenitor number is controlled through mesenchymal FGF receptor signaling” for further consideration at *eLife*. Your revised article has been favorably evaluated by Janet Rossant (Senior editor) and Tanya Whitfield (Reviewing editor). The manuscript is much improved and most of the comments have been addressed. There are some discrepancies with previously published work, but these have now been acknowledged and discussed.

A few remaining issues still need to be addressed before acceptance, as outlined below:

1) As requested, morphological data for the single mutants have been added for comparison in Figure 2, but no additional *Fgf9*^*-/-*^ single mutant analysis has been added to Figure 3 to show proliferation data for this genotype. Reference is also made to [38], where the authors state that the cochlear duct is of normal length and architecture in *Fgf9*^*-/-*^ mice. However, that paper has no quantitative data concerning proliferation either, and cochlear patterning is not analyzed in detail. If quantitative proliferation data on the single *Fgf9* KO to compare to the double KO are available, they should be included. If they are not available, the claim that the epithelial proliferation phenotype in the double mutant reflects a synergy between the two Fgfs should perhaps be toned down.

2) None of the data shown appear to use simple pairwise comparisons. Each graph usually shows one (or more) control situation and several experimental situations (e.g. Figure 3: one control and two experiments to which a statistical test is applied). If each experiment is compared with the same control, this is not a pairwise comparison, and ANOVA with multiple sample correction should be used. In any case, when an asterisk is used on the bar graphs, it should be clarified what is being compared with what, either with a description in the legend or by using a horizontal bar between the relevant control and the experimental case (e.g. Figure 2: are all samples being compared with the left hand (white) double heterozygote control?).

---

## [Author Response]

*As you will see, all three reviewers found the work interesting and potentially important. However, they felt that the data, as they stand, do not provide full support for the proposed model, and that further work and clarification is required. As a minimum, this should include a full characterisation of the mesenchymal phenotype in each of the mutants, including the* Fgf9 *and* Fgf20 *single mutants for comparison*.

We have added *Fgf9* and *Fgf20* single mutant data to Figure 2. The new data include representative images and quantitation of HC and SC numbers and cochlear length at E18.5.

*Improved quantitation of the sensory cell phenotype and levels of proliferation in epithelium and mesenchyme, ideally including a marker other than PHH3, will be important*.

We have labeled embryos with EdU at E11.5 and quantified cell proliferation in the Sox2^+^ domain. The genotypes analyzed include: *Fgf9*^*-/+*^;*Fgf20*βGal*/+*; *Fgf9*^*-/+*^*;Fgf20*βGal/βGal; *Fgf9*^*-/-*^;*Fgf20*βGal/βGal. These data are consistent with our previous data in Figure 3 showing reduced epithelial proliferation in the double mutant mouse embryo cochlear duct. The data is shown in Figure 3—figure supplement 1.

*Alternative interpretations of the mesenchymal phenotype should be considered and discussed*.

We have added to the Discussion additional interpretations of the mesenchymal phenotype.

*A fuller discussion of the phenotype caused by* Fgfr1 *KO in the epithelium is also required, since this also affects cochlear length*.

We have added to the Discussion additional comparisons of the epithelial phenotype to the phenotype seen in mice conditionally lacking FGFR1, published by the Ono et al.

*The revised version must include citation, acknowledgement and discussion of previously published work that impacts on the current study. In particular, it will be worth examining the IHC phenotype carefully here*.

We have added additional citations and acknowledgements as requested. In the genetic background used in our experiments we do not see any effects on IHC differentiation. The total number of IHCs is decreased in the *Fgf9/20* double knockout embryos and in the mesenchymal FGFR conditional knockouts in which the length of the cochlea is dramatically shortened. These points are now discussed in the revised manuscript.

*The full reviews are appended below*.

Reviewer #1:

*This is an interesting paper that builds on a previous study (*[18]*) demonstrating a role for* Fgf20 *in differentiation of the lateral compartment of the cochlea. The new information here is that FGF signalling from epithelium to mesenchyme is both required and sufficient for sensory progenitor proliferation and extension of the cochlear duct. The data are clear and support the conclusions well. However, there is no identification of a potential mechanism for the mesenchymal signal back to the cochlear duct that regulates its growth. As a result, the advance that the study makes over previous work appears to be somewhat incremental*.

We agree with the reviewers that we have not identified the mesenchymal signal back to the cochlear duct that regulates its growth. However, we have ruled out several candidate pathways. What we have done is identify that mesenchymal FGFR signaling, possibly through activation of *Etv4/5*, is a critical pathway that will regulate the putative mesenchymal signal back to the cochlear duct. This is an important key step in understanding the ultimate, and currently elusive, molecular mechanism that regulates cochlear progenitor cell growth.

*A few details of the manuscript could be clarified*:

*The normal expression patterns of* Fgfr1 *and* Fgfr2 *in the otic epithelium and periotic mesenchyme at the relevant stages should be presented or referred to early on in the manuscript.*

We have cited two references, for the *Fgfr1* and *Fgfr2* expression patterns in the otic epithelium and periotic mesenchyme at early, E10-12, stages of development. (Pirvola et al. J. Neuroscience 2000 and Neuron 2002).

Introduction: ‘…and that mesenchymal FGF signaling is sufficient to promote…’. This is ambiguous. Is this the same signalling from epithelium to mesenchyme as referred to in the previous sentence, or new signalling from mesenchyme to epithelium, also mediated by FGF?

We have clarified this stated conclusion by indicating that *Fgf9* and FGF20 signal to mesenchymal FGFRs and that mesenchymal FGFR signaling is necessary and sufficient to regulate cochlear progenitor cell number.

Figure 1
*and Results: Make sure the text follows the order of the panels in the figure and vice versa—i.e. show* Fgf20 *first in the figure*.

We have modified the text to follow the order of the figure.

Figure 1*: The sketch here does not appear to represent the expression patterns shown very well. The two FGF domains appear to abut each other directly, with a very sharp boundary at the junction between non-sensory (thinner) and sensory (thicker) epithelium. This is not shown in the sketch, which does not depict the full extent of the* Fgf20 *domain and shows a substantial gap between the two domains*.

We have redrawn the sketch to better reflect the expression patterns shown in the immunostained sections.

Figure 2*: It would be helpful to show the* Fgf20^-/-^; Fgf9^+/+^
*phenotype here, to illustrate the effects of the loss of* Fgf20 *alone, even though previously published*.

We have added the *Fgf9*^*-/-*^;*Fgf20*^*+/+*^ and *Fgf9*^*+/+*^;*Fgf20*^*-/-*^ genotypes in Figure 2 as requested. The *Fgf9*^*-/-*^;*Fgf20*^*+/+*^ genotype is comparable to the double heterozygous control and *Fgf9*^*-/-*^;*Fgf20*^*-/+*^ genotypes. The *Fgf9*^*+/+*^;*Fgf20*^*-/-*^ genotype is comparable to *Fgf9*^*-/+*^;*Fgf20*^*-/-*^ genotype. These data indicate that there is no cochlear patterning phenotype resulting from loss of *Fgf9* as long as one wild type allele of *Fgf20* is present. Furthermore, loss of one copy of *Fgf9* does not contribute to either outer hair cell loss or cochlear length shortening.

*In the subsection headed “*Fgf9 *and* Fgf20 *regulate cochlear length and* Fgf20*, not* Fgf9*, is required for lateral compartment differentiation and patterning”, when you state “…the normal pattern of three rows of OHCs… (*Figure 2*)”, there should be a reference to the later panels (C, D) here as well*.

We added reference to panels C and D as suggested.

Figure 2*: Please mark on the positions of IHC and OHC on these panels. It looks to me as if there are two rows of IHC in the* Fgf20^βGal/βGal^*, especially in the right hand panel. Please comment*.

We have marked IHC and OHC as suggested. Yes, there are two rows of IHC in the sensory epithelial clumps. This was previously reported by us.

*The density of each HC type is measured for several different genotypes. It should be clarified somewhere that where patterning of the organ of Corti is affected, the whole organ is fragmented and that cell measurements are (presumably) taken over patches where the sensory cells are still present. If density was measured along the whole length of the cochlea, presumably IHC density would also be affected*.

We counted three regions of each cochlea (at 10%, 40%, and 70% along the length). In total, more than 900µm of each cochlea was counted. Due to delayed hair cell differentiation in *Fgf20*^*-/-*^ compound mutants, we counted only the regions that contained differentiated sensory cells. Two rows of IHC are present in patch regions and no IHCs were detected in the region between the sensory patches. Therefore, IHC density was not changed throughout the cochlea. We explained this in the Methods section.

Figure 3*: Other areas marked by Jag1, but not highlighted, appear to show significant differences from controls, e.g. in the top left hand corner of the panel, the domain appears to be twice as big as in controls, whereas the stained area towards the bottom of the panel appears much reduced. Please comment*.

At E14.5, *Fgf9*^*-/-*^;*Fgf20*^*-/-*^ embryos had very shortened cochlear length compared to other genotypes. Therefore the angle of the cochlear turns in these embryos do not match that of other genotypes. The regions of Jag1 staining that are not boxed are sectioned at an oblique angle and thus cannot be compared. The boxed regions are examples of sections that perpendicularly transect the cochlear duct and thus can be compared across genotypes. This has been clarified in the figure legend.

*The Discussion is rather brief, and does not really exploit the detail provided in the Results. In particular, the Discussion does not cover the potential differences in roles between* Fgf9 *and* Fgf20*, and* Fgfr1 *and* Fgfr2*. My interpretation of the Results was that* Fgf20 *primarily signals through* Fgfr1 *in the cochlear epithelium to regulate patterning of lateral compartment of the organ of Corti, while* Fgf9 *primarily signals through* Fgfr2 *in the mesenchyme to regulate cochlear length, with some cross-activity and redundancy between the two FGF pathways. However, the Discussion merely focusses on the redundancy rather than any differences between the two. It would be interesting to relate any differences to the clear differences between the non-overlapping expression domains of each FGG in the developing cochlear duct, and to the expression domains of the FGFRs, which are not shown (see comment above)*.

We have included discussion regarding specific roles of *Fgf9* and *Fgf20* and their receptors as shown below (extracted from the Discussion section):

*“Fgf9* is expressed in non-sensory epithelia of the cochlea and loss of *Fgf9* results in defects in periotic mesenchymal cell proliferation […] FGFR1 was identified as the epithelial target receptor (18; 37).”

*In addition, there is no mention of how the mesenchyme may signal back to the epithelium to regulate cochlear growth. It is not made clear in the paragraph about RA signalling whether this is being proposed as the mechanism of mesenchymal to epithelial signalling, but the last sentence of the manuscript appears to indicate that this may be a second signalling pathway that is regulating mesenchymal cell behaviour*. *Do the authors have any speculation on how the mesenchyme may regulate cochlear duct growth? Are there any candidate factors?*

At present *Tbx1/Pou3f4* induction of RA signaling is the only known mesenchymal factor to induce sensory progenitor proliferation. We showed that *Tbx1* and *Pou3f4* were not changed in *Fgf9*/*Fgf20* mutant inner ear. However, we found that *Etv4* and *Etv5* were decreased in *Fgf9*/*Fgf20* mutant inner ear. We speculate that downstream targets of *Etv4* and *Etv5* might induce sensory epithelial proliferation. We described these possibilities in the Discussion.

Reviewer #2:

*One of my major criticisms of this paper is the failure to reconcile their data with data that is already published. A recent study of the* Fgfr1 *phenotype (*[33]*) has not been considered and needs to be evaluated in the context of the authors work. For example, in the subsection “Epithelial* Fgfr1 *but not* Fgfr2 *is required for lateral compartment differentiation”, the sentence “however, no quantitative data regarding affected cell types or cochlear length was reported” (in* Foxg1^Cre^ Fgfr1 *mutants) is untrue—the data is presented in Ono et al. Similarly, a paper showing the role of* Fgf20 *in the maintenance of Sox2*^*+*^
*progenitors (Munnamalai et al., 2012) has not been cited or considered*.

*With this in mind, it is probably worth highlighting the points that are inconsistent*:

*In* Fgf9/20 *knockouts,* Fgf20^Cre^;Fgfr1/2 *and* Dermo1^Cre^;Fgfr1/2 *mutants the inner hair cells are unaffected. In Ono et al., early deletion (using* Six1-Cre and Foxg1^Cre^*) of* Fgfr1 *affects IHC*;

We agree with the reviewer in that there are differences in the observed phenotypes between data shown here and that shown by [33]. We have modified the Results section to include the Ono et al. observations. In the Discussion, we attempt to reconcile these differences in phenotype. We think that the differences are most likely attributed to differences in genetic background, although the genetic background of the Ono et al. mice were not reported.

*Cochlear length is affected, quite significantly, in the early, epithelial deletions of* Fgfr1 *presented by Ono et al. Huh, Warchol and Ornitz do not find any reduction in epithelial size*;

We observed a 10-25% decrease of cochlear length in *Foxg1*^*Cre*^, *Fgfr1*^*-/f*^ mutants, which is comparable with what we observed in *Fgf20*^*-/-*^ mutants. We maintain our mouse lines on a 129X1/SvJ;C57BL/6J mixed background. We observed a shorter cochlear length in *Fgf20* mutant when on a pure C57BL/6J genetic background (not shown). We speculate that the difference between what we observed and Ono et al. may due to different genetic backgrounds.

*Proliferation of Sox2*^*+*^
*progenitors was unaffected in Ono's early deletions of* Fgfr1*.*

We observed that progenitor proliferation was decreased only with mesenchymal FGFR deletion (with *Dermo1*^*Cre*^) but not with epithelial FGFR deletion (with *foxg1*^*Cre*^ or *Fgf20*^*Cre*^). This result is consistent with the observations of Ono et al.

*Alternative interpretations of the mesenchymal requirement of FGFR signalling in the regulation of cochlear length need to be considered. One possibility is that the effect on cochlear length is secondary, not because of a direct action of* Fgf9/20 *on* Fgfr1 *and* 2 *in the mesenchyme, but because there is simply less mesenchyme. I think that it is quite likely that a reduction of* Fgfr1 *and* 2 *in the peri-otic mesenchyme at such early stages leads to a reduction in the amount of mesenchyme. As Doetzlhofer et al., had shown, the mesenchyme is important for outgrowth. The authors need to analyse the periotic mesenchyme in these mutants*.

Out data is consistent with the observations of Montcouquiol 2003 and Doetzlhofer 2004 showing that mesenchyme is important for epithelial progenitor growth, presumably through production of some trophic factor. Pirvola, 2004 showed that periotic mesenchyme proliferation was decreased in *Fgf9* mutant, resulting in a hypoplastic otic capsule. But this did not affect either cochlear length or progenitor proliferation (38). Therefore, decreased mesenchymal cell number is unlikely to be the direct cause of the epithelial phenotype. Rather, we propose that FGF signaling within mesenchyme affects some factor that signals reciprocally to epithelial progenitors. We have analyzed periotic mesenchyme for candidate signaling molecules and targets of FGFR signaling. We have shown that *Etv4* is decreased, but we have not yet found a mesenchymal-derived signaling molecule that could regulate epithelial proliferation.

*The correlation of sensory progenitors with cochlea length is quite a laboured point. It is clear that even in the absence of Sox2*^*+*^
*progenitors (see*
[22]*), the cochlea still extends. It is likely that progenitor proliferation plays at best a contributory role*.

We agree with the reviewer. For cochlea to elongate, at least two mechanisms are required, 1) Cochlear length should depend on the total number of available differentiated hair cells (which depend on total numbers of progenitor cells); 2) Cochlear length should depend on the proper organization of sensory cells into one row of IHC and 3 rows of OHCs, governed by convergent extension. Many PCP mutants that have disrupted convergent extension show decreased cochlear length. We posit that *Fgf9/20* signaling to mesenchyme is required for progenitor proliferation, thus affecting cochlear length. In addition, FGF20 signaling to epithelial FGFR1 regulates sensory cell differentiation and this also affects cochlear length, but, in our mouse genetic background, this is a relatively small effect (10-25%).

*The cochlear of* Fgf9 *nulls, as well as the* Fgf20 *nulls and* Fgf9/20 *double nulls should also be analysed. Are these the same as wild-types or is the* Fgf9 *and* Fgf20 *phenotype additive?*

We have added these data to Figure 2. *Fgf9*^*-/-*^ or *Fgf9*^*-/-*^;*Fgf20*^*-/+*^ does not affect cochlear length and HC/SC number. In addition, *Fgf20*^*-/-*^ and *Fgf9*^*-/+*^;*Fgf20*^*-/-*^ showed a similar phenotype.

Reviewer #3:

*In the first part of the paper, the authors show a role for* Fgf9 *on cochlear length; while* Fgf20 *seems to be affecting the number of prosensory progenitors that are being generated, or the way they are being patterned. This role for* Fgf20 *has been previously described by this group, and here they extend that work to show that* Fgf9 *does not play a role in this aspect. In*
Figure 2*, the paper suffers from not showing data from* Fgf20^-/-^
*mice alone (presumably because this is contained in their earlier publication). I think evidence from this mouse should be included here as part of the current analysis, so that direct comparison is possible; also, its absence makes the description of the various phenotypes rather confusing, and the authors might want to spend some time thinking of a way to describe this part more clearly*.

We have added the *Fgf9*^*-/-*^ and *Fgf20*^*-/-*^ data to Figure 2. See response to reviewer 2 above.

*In the end of this first section, the authors claim that there are synergistic effects between the two genes; but I am unclear about why they are making this interpretation, since the lack of any patterning defects in* Fgf9^-/*-*^*, seems to suggest that this FGF acts outside the prosensory domain to affect the length of the cochlear duct (*Figure 2*)*.

We agree with the reviewer that FGF9 acts outside of the prosensory domain to affect the length of cochlear duct. Our data shows that FGF20 also acts outside of the prosensory domain to affect the length of cochlear duct and functions together with FGF9. Receptor inactivation in mesenchyme strongly suggests that the target of FGF9 and FGF20 is periotic mesenchyme. Our conclusion is that there is synergism in signaling to mesenchyme, but no synergism in signaling to epithelium.

*The main point of the second section (*Figure 3*), is that* Fgf9 *plays a synergistic role with* Fgf20 *in the proliferation of the cells of the prosensory domain that will make up the organ of Corti. This is an interesting observation, but seems to contradict the separate roles that the two genes play as described in*
Figure 2*, namely prosensory (*Fgf20*) and non-prosensory (*Fgf9*)*.

The reviewer has astutely identified the key observation of this work. In Figure 2 the phenotype is observed at E18.5. We are hypothesizing, based on the data shown, that the FGF9/20 double knockout encompasses two phenotypes that are developmentally distinct and separable. 1) A patterning phenotype that is due to FGF20 signaling to epithelial FGFR1 as was described in our previous publication (Huh, 2012). This patterning phenotype is independent of FGF9. 2) A cochlear length phenotype that demonstrates synergism between FGF9 and FGF20. Figure 3 begins to define how these two phenotypes differ. The key data is in Figure 3. This shows that Control or FGF20 knockout cochlea have normal numbers of sensory progenitor cells. In contrast the *Fgf9/20* double knockout has significantly reduced numbers of progenitor cells. From these data we conclude that FGF20 alone does not affect progenitor cell number, but rather functions later in development to regulate epithelial patterning (differentiation). In contrast FGF9/20 together regulate progenitor cell number. Therefore, these are really not contradictory results and actually define two distinct developmental processes.

*The data here are also a little difficult. In*
Figure 3*, they are counting cells in histological sections rather than whole mounts? But it is difficult to know how much of the prosensory domain they actually measure, and did they count in exactly the same basal-apical position? This may have been difficult to do, since the (*Fgf20^-/-^::Fgf9^-/+^) *does show a change in patterning (“less compact”). Thus, the modest counting difference that they observe in*
Figure 3
*could be due to differences in cell distribution within the duct? Indeed, from this figure there does seem to be a trend in their data showing that* Fgf20 *has an effect on its own, without* Fgf9*, even if their statistical test does not show a significant difference; and so* Fgf9 *could be skewing the data by changing the shape of the duct? This leads me to be skeptical of their interpretation that the genes are acting together, instead of independently on two different elements of the observed phenotype*.

We counted pHH3^+^ cells within the Sox2^+^ sensory progenitor domain throughout the entire cochlear duct, counting every second section from base to apex. In the revised manuscript, we have repeated this analysis for the E11.5 time point using EdU labeling. The result was consistent with the data in Figure 3 shown in Figure 3—figure supplement 1.

Figure 5*: I don't think the conclusion that “together, these data show that FGFR1/2, expressed in mesenchymal cells, are required for proper cochlear length formation and for sensory progenitor proliferation, but not for cochlear pattern formation or differentiation” is completely supported in the case of proliferation, since they fail to isolate the two genes for the proliferation part of the study (*Figure 5*), and only look at the double KO. How do we know that FGFR1 is not responsible*, *on its own, for the proliferation defect, an observation that would be consistent with the previous Pirvola study, I think?*

The reviewer is correct. We have reworded the paragraph to state that mesenchymal FGFR signaling is a necessary determinant of cochlear length and sensory progenitor proliferation, but not for cochlear pattern formation or differentiation.

The Pirvola study inactivated epithelial *Fgfr1* but did not measure proliferation or effects on cochlear length. We repeated this experiment and showed that there is no effect on sensory progenitor proliferation (Figure 4—figure supplement 1) and a very small effect on cochlear length (9%).

Figure 6*: In which they show that activation of FGFR signalling is sufficient to affect prosensory proliferation, is very interesting, and complements the KO data*.

*Discussion*:

*The authors state that their evidence shows that “in this study, we found that epithelial FGF9 and FGF20 signaling to mesenchymal FGFR1 and FGFR2 is required for normal levels of cochlear sensory progenitor proliferation and that inactivation of either the ligands or the mesenchymal receptors results in a shortened cochlea*.*”*

*Also, that “…the expression patterns of these two FGFs do not overlap*, *but nevertheless they appear to signal to a common target tissue, periotic mesenchyme in the developing inner ear…”*

*But how do we know that pro-sensory-derived FGF9 and FGF20 are signaling through mesenchymal FGFR1/2? Why couldn't they also be signaling through epithelial FGFRs? The fact that overstimulation of FGFR1/2 in mesenchyme changes the epithelial proliferation may suggest this is one route, but does not prove this, I don't think*.

We posit that FGF9/20 are unlikely to signal to epithelial FGFRs to regulate progenitor cell proliferation based on phenotypic comparisons of the different tissue-specific receptor knockouts. Epithelial deletion of *Fgfr1/2* did not show a dramatic decrease in cochlear length or progenitor proliferation compared to *Fgf9/20* deletion. On the other hand, mesenchymal deletion of *Fgfr1/2* showed a severe decrease of cochlear length and progenitor proliferation, which was comparable to the *Fgf9/20* mutant, but no patterning phenotype. We believe that these data demonstrate that mesenchymal *Fgfr1/2* are required for proper cochlear length formation and that this phenotype likely results from reduced progenitor proliferation and fewer progenitor cells. We have elaborated on this in the Discussion.

[Editors' note: further revisions were requested prior to acceptance, as described below.]

*1) As requested, morphological data for the single mutants have been added for comparison in*
Figure 2*, but no additional* Fgf9^-/-^
*single mutant analysis has been added to*
Figure 3
*to show proliferation data for this genotype. Reference is also made to*
[38]*, where the authors state that the cochlear duct is of normal length and architecture in* Fgf9^-/-^
*mice. However, that paper has no quantitative data concerning proliferation either, and cochlear patterning is not analyzed in detail. If quantitative proliferation data on the single* Fgf9 *KO to compare to the double KO are available, they should be included. If they are not available, the claim that the epithelial proliferation phenotype in the double mutant reflects a synergy between the two Fgfs should perhaps be toned down.*

We have not measured proliferation in *Fgf9*^*-/-*^ embryos because we have not observed any phenotype that affects hair or supporting cell number or overall length of the cochlear duct in these mice. If the editors or reviewers require these data we can add it, but this will require 3-4 weeks to time mate and analyze the appropriate mice.

We have modified the text to say that comparison of cochlear epithelial cell proliferation was compared to double heterozygous controls.

We do not use the word synergy, but we do state that FGF9 and FGF20 have redundant function. This is based on the lack of a cochlear phenotype in *Fgf9*^*-/-*^ embryos and the appearance of a sensory epithelial proliferation phenotype in the *Fgf9/20* double mutant that is not observed in *Fgf20*^*-/-*^ cochleae.

*2) None of the data shown appear to use simple pairwise comparisons. Each graph usually shows one (or more) control situation and several experimental situations (e.g.*
Figure 3*: one control and two experiments to which a statistical test is applied). If each experiment is compared with the same control, this is not a pairwise comparison, and ANOVA with multiple sample correction should be used. In any case, when an asterisk is used on the bar graphs, it should be clarified what is being compared with what, either with a description in the legend or by using a horizontal bar between the relevant control and the experimental case (e.g.*
Figure 2*: are all samples being compared with the left hand (white) double heterozygote control?)*.

We have clarified the statistical comparisons to state that two-way comparisons used Student’s t test and we used one-way ANOVA for comparison of more than three samples.